# A mechanism of cohesin-dependent loop extrusion organizes zygotic genome architecture

Johanna Gassler[1,†], Hugo B Brandão[2,†] (iD), Maxim Imakaev[3,4], Ilya M Flyamer[5], Sabrina Ladstätter[1], Wendy A Bickmore[5], Jan-Michael Peters[6] (iD), Leonid A Mirny[2,3,*] (iD) & Kikuë Tachibana[1,**] (iD)

## Abstract

Fertilization triggers assembly of higher-order chromatin structure from a condensed maternal and a naïve paternal genome to generate a totipotent embryo. Chromatin loops and domains have been detected in mouse zygotes by single-nucleus Hi-C (snHi-C), but not bulk Hi-C. It is therefore unclear when and how embryonic chromatin conformations are assembled. Here, we investigated whether a mechanism of cohesin-dependent loop extrusion generates higher-order chromatin structures within the one-cell embryo. Using snHi-C of mouse knockout embryos, we demonstrate that the zygotic genome folds into loops and domains that critically depend on Scc1-cohesin and that are regulated in size and linear density by Wapl. Remarkably, we discovered distinct effects on maternal and paternal chromatin loop sizes, likely reflecting differences in loop extrusion dynamics and epigenetic reprogramming. Dynamic polymer models of chromosomes reproduce changes in snHi-C, suggesting a mechanism where cohesin locally compacts chromatin by active loop extrusion, whose processivity is controlled by Wapl. Our simulations and experimental data provide evidence that cohesin-dependent loop extrusion organizes mammalian genomes over multiple scales from the one-cell embryo onward.

**Keywords** chromatin structure; cohesin; loop extrusion; reprogramming; zygote
**Subject Categories** Cell Cycle; Chromatin, Epigenetics, Genomics & Functional Genomics; Development & Differentiation
**The EMBO Journal (2017) 36: 3600–3618**

See also: **G Wutz et al** (December 2017) and **JHI Haarhuis & BD Rowland** (December 2017)

## Introduction

Chromatin is assembled and reprogrammed to totipotency in the one-cell zygote that has the potential to generate a new organism. The chromatin template upon which higher-order structure is built in the embryo is different for the maternal and paternal genomes at the time of fertilization. The maternal genome is inherited from the meiosis II egg in which chromosomes are condensed in a mitotic-like state. In contrast, the paternal genome is contributed from compacted sperm chromatin that is extensively remodeled upon fertilization, as protamines are evicted and naïve nucleosomal chromatin is established (Rodman et al, 1981). The two genomes are reprogrammed as separate nuclei with distinct epigenetic signatures at the zygote stage (Mayer et al, 2000; Oswald et al, 2000; van der Heijden et al, 2005; Torres-Padilla et al, 2006; Ladstätter & Tachibana-Konwalski, 2016). With the exception of imprinted loci, differences in chromatin states are presumably eventually equalized to facilitate the major zygotic genome activation (ZGA), which occurs at the two-cell stage in mice (Aoki et al, 1997; Hamatani et al, 2004; Inoue et al, 2017). The establishment of zygotic genome architecture is therefore likely important for transcriptional onset and embryonic development.

Higher-order chromatin structures including chromatin loops, topologically associating domains (TADs), and compartmentalization of active and inactive chromatin are established during embryonic development (Du et al, 2017; Flyamer et al, 2017; Hug et al, 2017; Ke et al, 2017). Using single-nucleus high-resolution chromosome conformation capture (snHi-C), we previously identified the presence of loops and TADs in mouse zygotes and oocytes (Flyamer et al, 2017) by averaging contact maps over the positions of annotated TADs and loops (Rao et al, 2014). In contrast, bulk Hi-C of mouse zygotes detected only weak or obscure domain structures that strengthened during preimplantation development (Du et al, 2017; Ke et al, 2017). However, it is not clear whether these bulk Hi-C approaches would detect the TADs and loops that

1  Institute of Molecular Biotechnology of the Austrian Academy of Sciences (IMBA), Vienna Biocenter (VBC), Vienna, Austria
2  Harvard Graduate Program in Biophysics, Harvard University, Cambridge, MA, USA
3  Institute for Medical Engineering and Science, Massachusetts Institute of Technology (MIT), Cambridge, MA, USA
4  Department of Physics, Massachusetts Institute of Technology (MIT), Cambridge, MA, USA
5  MRC Human Genetics Unit, Institute of Genetics and Molecular Medicine, University of Edinburgh, Edinburgh, UK
6  Research Institute of Molecular Pathology (IMP), Vienna Biocenter (VBC), Vienna, Austria
   *Corresponding author. Tel: +1 617 452 4862; E-mail: leonid@mit.edu
   **Corresponding author. Tel: +43 1 79044 4670; E-mail: kikue.tachibana@imba.oeaw.ac.at
   †These authors contributed equally to this work

  

are expected to form in interphase germinal vesicle-stage meiosis I oocytes (Flyamer *et al*, 2017). A combination of biological and technical factors, including smaller cell numbers used to analyze zygotes compared to blastocysts and different analyses of TAD aggregation data, may limit the detection of higher-order chromatin structures by bulk Hi-C (Du *et al*, 2017; Ke *et al*, 2017). Interestingly, TADs or loops are not detected in the rapidly dividing nuclei in early *Drosophila* embryos (Hug *et al*, 2017), or in metaphase II oocytes with condensed chromosomes (Du *et al*, 2017; Ke *et al*, 2017). Mitotic chromosomes in HeLa cells also lack TADs and loops, suggesting that this feature is not specific to meiosis II oocytes (Naumova *et al*, 2013). Therefore, which higher-order chromatin structures are assembled in mammalian zygotes remains unresolved and the mechanisms that establish these structures in embryos are not known.

Studies in other cell types are beginning to provide insights into possible mechanisms that lead to the establishment of higher-order chromatin structures. An early stepping-stone toward understanding chromatin structure was the unexpected finding that the cohesin complex, known to be essential for sister chromatid cohesion, is expressed in post-mitotic cells (Wendt *et al*, 2008). Cohesin is a tripartite ring consisting of Scc1-Smc3-Smc1. The cohesin ring is loaded onto chromatin by a loading complex composed of Nipbl/Scc2 and Mau2/Scc4 and is released from chromosomes by Wapl (Ciosk *et al*, 2000; Gandhi *et al*, 2006; Kueng *et al*, 2006; Tedeschi *et al*, 2013). Mutations in Nipbl cause Cornelia de Lange syndrome (CdLS), which is characterized by gene expression defects and altered chromatin compaction but no obvious defects in sister chromatid cohesion (Krantz *et al*, 2004; Tonkin *et al*, 2004; Musio *et al*, 2006; Deardorff *et al*, 2007; Nolen *et al*, 2013). Therefore, the idea emerged that cohesin may have roles beyond holding sister chromatids together. The discovery that cohesin colocalizes with CTCF and mediates its transcriptional insulation led to the conceptual advance that cohesin may hold DNA together not only between sister chromosomes but also in *cis*, within chromatids (Parelho *et al*, 2008; Wendt *et al*, 2008). This is supported by the finding that depletion of Wapl leads to an increased residence time of chromosome-bound cohesin; moreover, it causes the formation of prophase-like chromosomes with cohesin-enriched axial structures termed "vermicelli" in G0 cells and affects chromosome condensation (Lopez-Serra *et al*, 2013; Tedeschi *et al*, 2013). This discovery suggested that cohesin organizes intra-chromatid loops in interphase.

Chromosome conformation capture (3C)-based methods described interphase TAD structures with cohesin and CTCF enrichment at the boundaries (Dixon *et al*, 2012; Nora *et al*, 2012; Rao *et al*, 2014; Vietri Rudan *et al*, 2015). These observations led to the testable prediction that cohesin is required for TAD formation. Cohesin depletion approaches including HRV protease-mediated cleavage, siRNA knockdown, or conditional genetic knockout in cycling and differentiated cells had only minor effects on chromatin structure (Seitan *et al*, 2013; Sofueva *et al*, 2013; Zuin *et al*, 2014), suggesting either that cohesin is not essential for TAD formation or that protein depletion was inefficient. However, it was recently shown that auxin-inducible cohesin degradation leads to loss of TADs and loops in cancer cell lines (Rao *et al*, 2017; Wutz *et al*, 2017). Genetically knocking out the cohesin loading complex subunits Nipbl in post-mitotic liver cells

and Mau2 in HAP1 cells also diminished the strength of TADs and loops (Haarhuis *et al*, 2017; Schwarzer *et al*, 2017).

A mechanism explaining the formation of TADs and loops is provided by the loop extrusion model. In this model (Sanborn *et al*, 2015; Fudenberg *et al*, 2016), dynamic chromatin loops are created in *cis* by loop-extruding factors (LEFs). When a LEF binds to chromatin, it starts to translocate along the fiber in both directions, connecting successively further points, thus extruding a loop (Fig 1A). Translocation of loop extruders is hindered by boundary elements often located at TAD boundaries. Individual extruded loops are stochastic and can neither be visible in population Hi-C nor distinguished from other contacts in snHi-C. Loop extrusion, however, leads to enrichment of contacts within TADs and recapitulates peaks of contact frequency commonly referred to as loops (Fig 1B). Cohesin is hypothesized to act as a loop extruder in interphase, while CTCF is likely the most prominent boundary element in mammalian cells (Sanborn *et al*, 2015; Fudenberg *et al*, 2016; Hansen *et al*, 2017; Nora *et al*, 2017).

Here, we provide evidence that cohesin-dependent loop extrusion organizes higher-order chromatin structures of mammalian zygotic genomes. We show that cohesin is essential for chromatin loops and TADs but not compartments and other large-scale zygote-specific structures in one-cell embryos. We find that inactivating

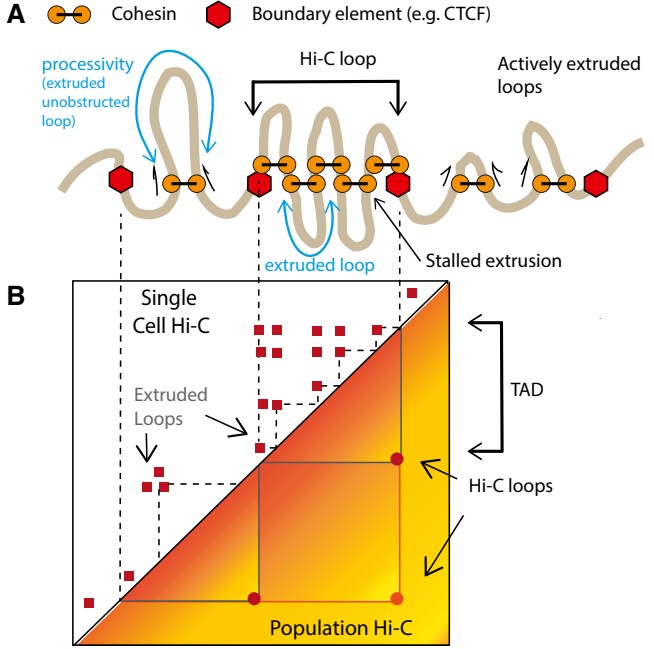

**Figure 1.  Relationship between single-cell and population Hi-C maps in light of the loop extrusion model.**

A   A schematic illustration for the loop extrusion mechanism. The model posits that cohesin (the LEF) processively extrudes chromatin loops and is hindered by other cohesins or boundary elements such as CTCF.

B   We illustrate the distinction between cohesin-extruded loops which result in variable contacts in single-cell maps and Hi-C loops which represent a population-average picture of extruded loops stalled at boundary elements. TADs in population Hi-C maps are generated by cohesin-extruded loops.

cohesin release by Wapl depletion exacerbates differences in loop strengths between the maternal and paternal genomes that may be related to reprogramming. Remarkably, simulations indicate that most differences in global organization between the two zygotic genomes can be driven by changes in cohesin density and loop extrusion processivity. We further discovered that cohesin limits inter-chromosomal interactions by compacting chromatin; simulations indicate that this effect is due to altering the effective surface of chromosomes. We propose that cohesin-dependent loop extrusion organizes chromatin at multiple genomic scales from the mammalian one-cell embryo onward.

## Results

### Loops, TADs, and compartments are formed as early as in one-cell embryos

Using snHi-C, we recently found that mouse zygotic genomes are organized into chromatin loops, TADs, and compartments as early as G1 phase (Flyamer *et al*, 2017; Fig 2A). However, bulk Hi-C of zygotes detected few or obscure TAD structures until around the eight-cell stage (Du *et al*, 2017; Ke *et al*, 2017). To attempt to resolve this conflict, we re-analyzed these recent data (Du *et al*, 2017; Ke *et al*, 2017).

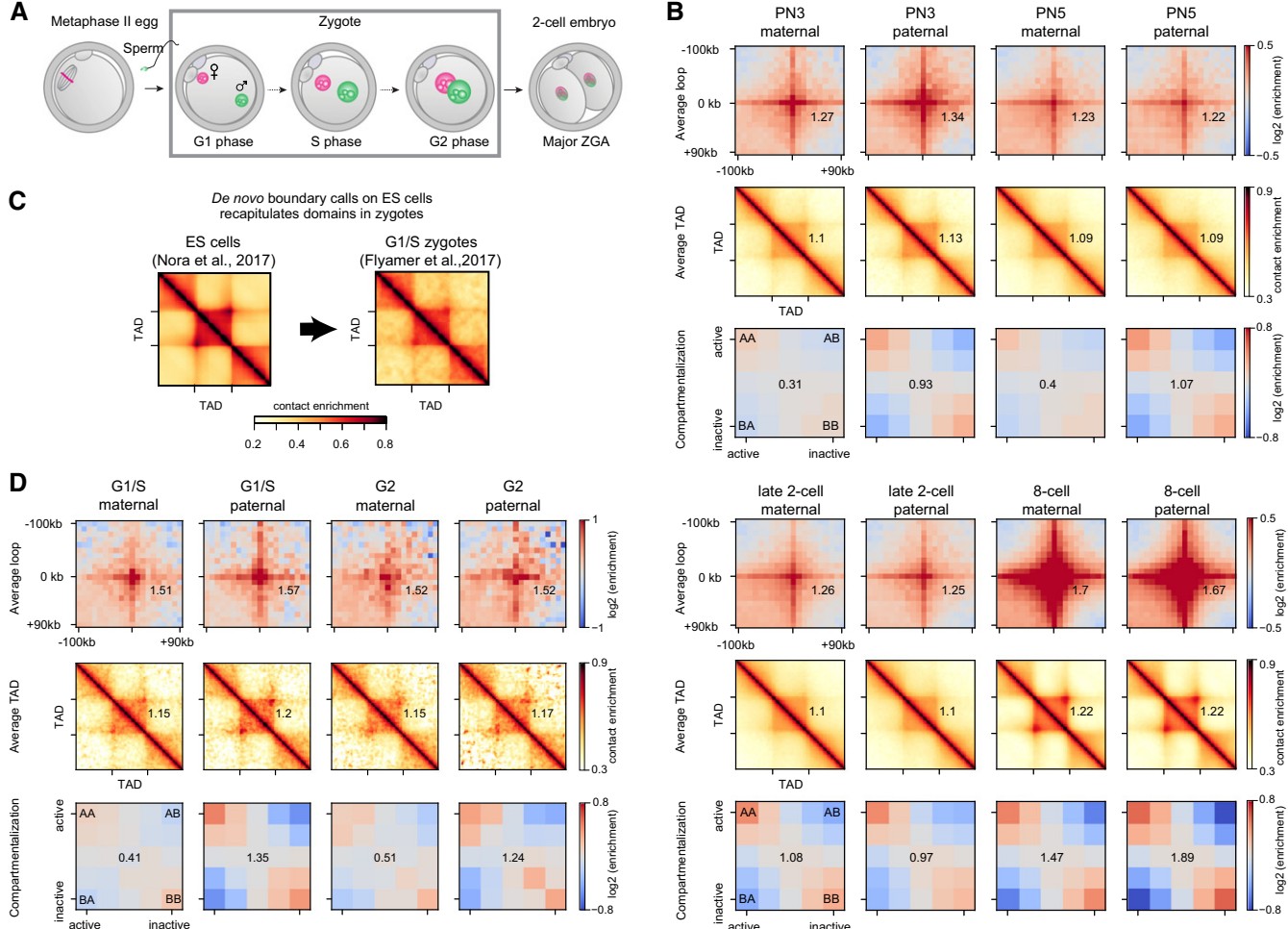

**Figure 2.   Zygotic chromatin is organized into loops, TADs, and compartments that change during the first cell cycle.**

A   Embryonic development from fertilization of the metaphase II egg by sperm, to zygote formation and division, to the two-cell embryo. Maternal and paternal genomes form separate nuclei in the zygote. The major zygotic genome activation (ZGA) occurs in the two-cell mouse embryo.

B   Average chromatin loops, TADs, and compartments are detectable in maternal and paternal chromatin from the one-cell embryo onward; data re-analyzed from Du *et al* (2017). Zygotic pronuclear stage 3 (PN3) and stage 5 correspond to S phase and G2 phase, respectively. The average strength of each feature is shown inset into each corresponding panel (see Materials and Methods).

C   We *de novo* annotated TAD boundaries (see Materials and Methods) in mouse ES cells (Nora *et al*, 2017) and show that TADs in wild-type zygotes are detected (Flyamer *et al*, 2017). To further verify that TAD detection in zygotes is insensitive to the choice of annotated boundaries, see Fig EV1.

D   The strength of average loops, TADs, and compartments becomes more similar between the maternal and paternal genomes as the zygotic cell cycle progresses (G1/S: Flyamer *et al*, 2017; G2: this work; *n*(maternal) = 18 and *n*(paternal) = 13 nuclei, based on two independent experiments using five and six females). The average strength of each feature is shown inset into each corresponding panel (see Materials and Methods).

Source data are available online for this figure.

Loop and TAD locations are generally conserved across cell types (Dixon *et al*, 2012; Rao *et al*, 2014) but are unknown in zygotes. Therefore, to uncover higher-order chromatin organization in zygotes, we used a list of loop loci identified in CH12-LX cells (Rao *et al*, 2014). For Hi-C data on low numbers of cells (Ulianov *et al*, 2017), loops and TADs are most visible when averaged over multiple positions (Flyamer *et al*, 2017) and normalized relative to control regions that are selected from random shifts of loop loci (Appendix Fig S1A). Using our approach, we found that these data support the presence of loops and TADs in eight-cell, two-cell, and even one-cell embryos (Fig 2B; Appendix Fig S1B) and are in agreement with previous findings that TADs and loops become stronger with progressing development (Du *et al*, 2017; Ke *et al*, 2017). To exclude that these results are biased toward TADs called in CH12-LX cells (Rao *et al*, 2014), we extended the analysis to include TADs called *de novo* in a variety of cell types including ES cells (Nora *et al*, 2017; Figs 2C and EV1). We found that all *de novo* TAD calls, on over 15 data sets and multiple cell types, resulted in contact enrichments in all of the wild-type zygote data sets (Figs 2C and EV1; this work, Du *et al*, 2017; Flyamer *et al*, 2017; Ke *et al*, 2017). Notably, contact enrichments were absent in metaphase II oocytes (Du *et al*, 2017), which, like mitotic cells, harbor condensed chromosomes (see Fig 2A) that presumably lack TADs (Naumova *et al*, 2013). Further, we discovered that zygotes lacking cohesin also do not form contact enrichments (see below).

In addition to this aggregate averaging analysis, we have visually identified certain genomic regions with TAD structures in heatmaps of bulk Hi-C zygote data (Fig EV2; Du *et al*, 2017; Ke *et al*, 2017), suggesting that this organization can be detected independently of aggregate analysis. Together, these findings strongly support the folding of zygotic genomes into higher-order chromatin structures.

### Zygotic genome architecture changes during the first cell cycle

Higher-order zygotic chromatin structure is established *de novo* for paternal chromatin and re-established after chromosome decondensation for the maternal genome. We noted that loops visually differed in strength between the parental genomes in G1 phase, with stronger loops seen in paternal chromatin (Fig 2B and D). However, these were not significantly different using a conservative statistical test for differences in loop strength ($P = 0.28$ with Flyamer *et al* (2017) G1/S data, and $P = 0.34$ with Du *et al* (2017) PN3 data; permutation test; Materials and Methods). It is conceivable that loops, TADs, and compartments change during the first cell cycle. To test this, we performed snHi-C of nuclei isolated from G2-phase zygotes (Fig 2D; see also Tables EV1 and EV2). We found that zygotic genomes are organized into TADs, loops, and compartments in G2 (Fig 2D), like in G1 phase. However, average loop and TAD strengths had further equalized between the parental genomes by G2 phase (Fig 2B and D; $P = 0.88$ with our G2 data and $P = 0.62$ with Du *et al* (2017) PN5 data; permutation test) and were not significantly different from G1 ($P > 0.055$, by the permutation test). To probe loops on a finer scale, we separated them into small (100–150 kb), intermediate (150–250 kb), and large (250–500 kb) and computed average loops for each distance. We found that paternal chromatin has higher contact frequency than maternal primarily for small- and intermediate-length loops in G1 (Appendix Fig S1C; $P < 0.05$, bootstrapping), which could be a consequence of loop

formation following protamine–histone exchange on sperm chromatin.

Likewise, compartment strengths differ between the maternal and paternal genomes in G1/S phase (Fig 2B and D), with maternal being much weaker and almost absent. In contrast to average loop and TAD strengths, a difference between maternal and paternal compartmentalization persisted through G2 (Fig 2B and D), consistent with recent reports (Du *et al*, 2017; Ke *et al*, 2017). We thus conclude that any initial differences in loop and TADs between zygotic maternal and paternal genomes become less evident by the end of the first cell cycle.

### Cohesin is essential for zygotic chromatin folding into loops and domains

To gain insights into the mechanisms that generate zygotic genome architecture, we tested whether the candidate loop-extruding factor cohesin is required for the formation or maintenance of loops and domains. We used a genetic knockout approach based on *(Tg)Zp3*-Cre, which conditionally deletes floxed alleles during the weeks of oocyte growth and generates maternal knockout zygotes after fertilization (Fig 3A). We have previously shown that Scc1 protein is efficiently depleted and sister chromatid cohesion fails to be established in $Scc1^{\Delta(m)/+(p)}$ zygotes (hereafter referred to as $Scc1^{\Delta}$ according to the maternal allele; see Fig EV3B and Ladstätter & Tachibana-Konwalski, 2016). Since sister chromatid cohesion is maintained by Rec8-cohesin in oocytes (Tachibana-Konwalski *et al*, 2010; Burkhardt *et al*, 2016), Scc1 depletion has no effect on chromosome segregation prior to fertilization, and therefore, a clean Scc1-cohesin knockout zygote is generated.

To test whether Scc1 is essential for TADs and loops in zygotes, we performed snHi-C (Flyamer *et al*, 2017) on genetically modified embryos. Both chromatin structures were detectable in control $Scc1^{fl}$ zygotes (Fig 3B). Remarkably, TADs and loops were largely, if not entirely, absent in $Scc1^{\Delta}$ zygotes, in both maternal and paternal nuclei (Figs 3B and EV3C). In contrast, compartmentalization of active and inactive chromatin from both maternal and paternal genomes was increased over 1.8-fold in $Scc1^{\Delta}$ compared to controls (Fig EV3C). We conclude that cohesin is essential for loops and domains and antagonizes compartmentalization, consistent with the notion that independent and possibly competing mechanisms generate these higher-order chromatin structures (Haarhuis *et al*, 2017; Nora *et al*, 2017; Nuebler *et al*, 2017; Schwarzer *et al*, 2017; Wutz *et al*, 2017).

### Wapl controls the size of cohesin-dependent chromatin loops

The loss of loops and domains in the absence of cohesin could either be due to an indirect effect, for example, on gene expression, or reflect a direct requirement for cohesin in loop formation or maintenance. The loop extrusion model predicts that increasing the residence time of cohesin on chromosomes strengthens existing loops and promotes the formation of longer loops in a population of cells (Fudenberg *et al*, 2016). The residence time of cohesin on chromatin can be increased more than 10-fold by inactivating cohesin release through Wapl depletion (Tedeschi *et al*, 2013). To test whether TADs and loops in zygotes are enhanced by

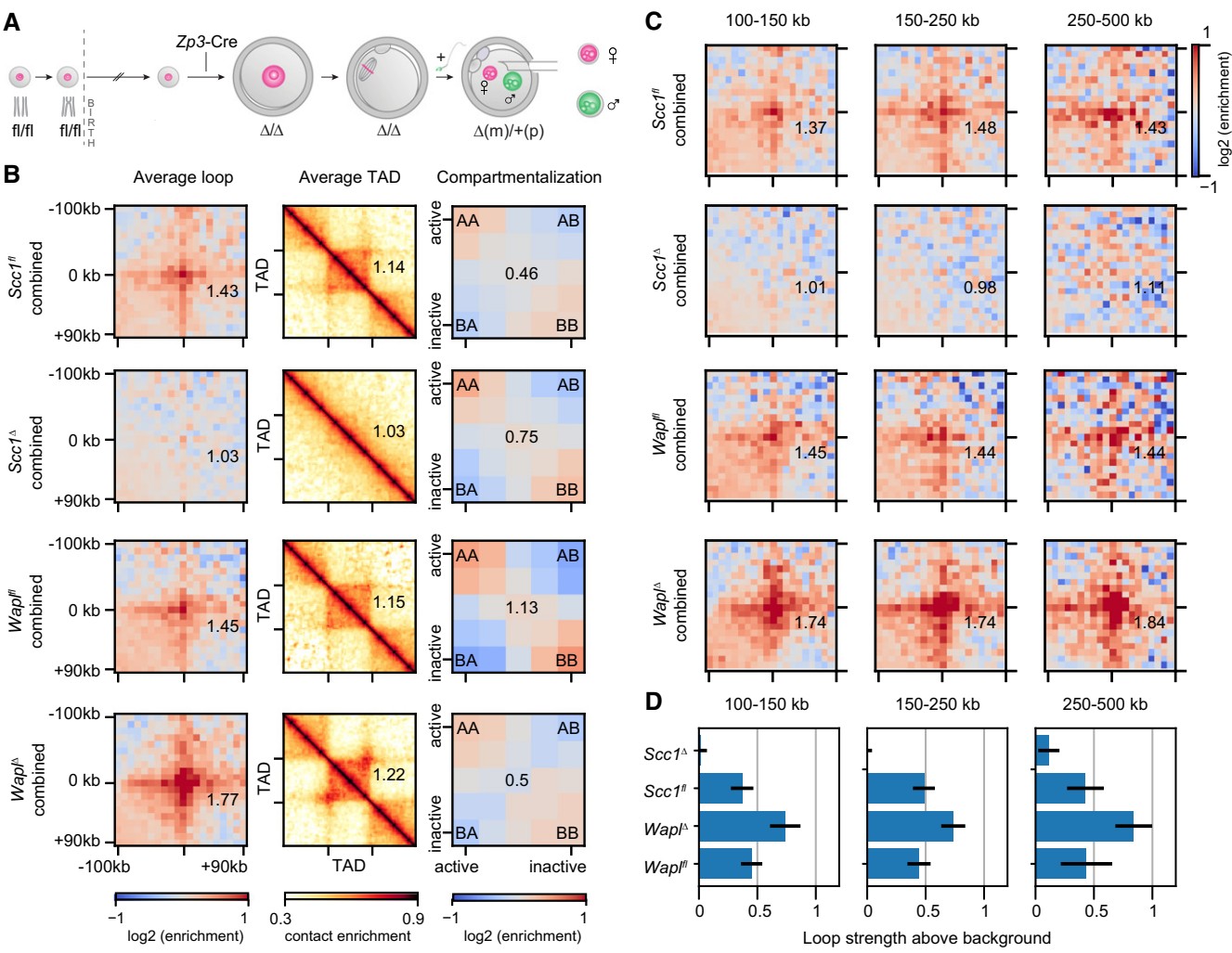

**Figure 3.  Conditional genetic knockouts of *Scc1* and *Wapl* reveal cohesin's essential role in the formation of loops and TADs in mouse zygotes.**

A   Generation of conditional genetic knockout oocytes by *Zp3*-Cre recombinase in post-recombination growing-phase mouse oocytes. Fertilization produces maternal knockout zygotes (maternal (m) and paternal (p) alleles). Maternal and paternal nuclei are extracted from zygotes before being subjected separately to snHi-C.

B   Average loops, TADs, and compartments in control (*Wapl^fl* and *Scc1^fl*), *Scc1^Δ*, and *Wapl^Δ* zygotes. Both maternal and paternal data are shown pooled together. Data are based on $n(Wapl^{fl}) = 17$, $n(Wapl^{\Delta}) = 17$, $n(Scc1^{fl}) = 30$, and $n(Scc1^{\Delta}) = 45$ nuclei, from at least two independent experiments using two to three females per genotype each.

C   Average loops, separated by size for control, *Scc1^Δ*, and *Wapl^Δ* zygotes for maternal and paternal data pooled together.

D   Loop strengths for heatmaps above, defined as the fractional enrichment above background levels (see Materials and Methods). Error bars displayed are the 95% confidence intervals obtained by bootstrapping pooled single-cell loops.

Source data are available online for this figure.

inactivating release of chromosomal cohesin, we generated *Wapl^Δ* ^{(m)/ + (p)} (*Wapl^Δ*) zygotes using the same strategy as described for *Scc1* (Fig 3A). The genetic deletion efficiency of *Wapl* is > 98% ($n = 85$ mice) (M. da Silva, J. M. Peters, personal communication), though we could not quantify the extent of protein depletion due to lack of Wapl antibodies that recognize the endogenous protein by immunofluorescence. We performed snHi-C of S/G2-phase *Wapl^Δ* zygotes and compared these to control data from *Wapl^fl* zygotes, which are wild-type for *Wapl*. Both TADs and loops were stronger in *Wapl^Δ* compared to control zygotes (Figs 3B and EV3C; see also Tables EV1 and EV2), in agreement with what has been observed in *Wapl^Δ* HAP1 and Wapl RNAi HeLa cells (Haarhuis *et al*, 2017; Wutz *et al*, 2017). Although formally we cannot exclude that these effects

are due to changes in gene expression, the most parsimonious explanation is that the effect of cohesin is direct; this accounts for the fact that cohesin depletion results in loss of TADs and loops, and increasing cohesin residence time by Wapl depletion results in stronger TADs and loops. Consistent with this, Nipbl depletion leads to loss of TADs and loops irrespective of changes in gene expression (Schwarzer *et al*, 2017). We conclude that cohesin release from chromosomes by Wapl is essential for regulating TADs and other local chromatin structures.

In addition to an effect on loops and TADs, we also observe that in the absence of Wapl, compartments became weaker than in controls by over 1.7-fold in both paternal and maternal genomes (Figs 3A and EV3C). These observations lend further support to the

idea that cohesin antagonizes compartmentalization, and are consistent with data and simulations in recent work (Haarhuis *et al*, 2017; Nuebler *et al*, 2017).

We next tested whether inactivating cohesin release from chromosomes causes changes to average strengths of loops. We found that loops are stronger in pooled *Wapl*$^\Delta$ zygote data compared to controls for all tested genomic distances (Fig 3C and D; *P* < 0.05, by bootstrapping). Interestingly, unlike for controls in which loop strength was invariant with increasing distance, *Wapl*$^\Delta$ zygotes displayed increasing loop strength from short to large distances with up to 80% enrichment of contacts above background levels (Fig 3D). These results are consistent with the loop extrusion mechanism and suggest that in wild-type cells, Wapl limits the extent of loop extrusion by releasing cohesin from chromosomes, impeding the amount of chromatin-associated cohesin and its processivity. Altogether, we conclude that cohesin directly regulates loop and domain formation or maintenance in the one-cell embryo.

### Cohesin organizes chromosomes at the sub-megabase scale

To further investigate how cohesin shapes genome architecture, we studied the genome-wide contact probability, $P_c(s)$, for chromatin loci separated by genomic distances, s. Consistent with our previous observations of wild-type zygotes (Flyamer *et al*, 2017), control cells have a $P_c(s)$ curve that changes slowly below 500 kb, reflecting local chromatin compaction; it changes steeply at or after 500 kb in both maternal and paternal chromatin and exhibits another plateau near 10 Mb in maternal chromatin, likely reflecting long-range chromatin interactions remaining from compaction to the mitotic state (Fig 4A; Appendix Fig S2; see also Tables EV1 and EV2; Naumova *et al*, 2013; Flyamer *et al*, 2017). Interestingly, the $P_c(s)$ curve of *Scc1*$^\Delta$ zygotes lost the shallow < 1 Mb region and followed a power law of $s^{-1.5}$, up to 1 Mb in both maternal and paternal genomes; the power law stretched up to 10 Mb in paternal chromatin (Fig 4B; Appendix Fig S2). This indicates that in the absence of cohesin, zygotic chromatin resembles a three-dimensional random walk as previously observed in yeast (Tjong *et al*, 2012; Halverson *et al*, 2014; Mizuguchi *et al*, 2014). Conversely, in *Wapl*$^\Delta$ zygotes, the contact probability was enriched and more shallow up to ~300 kb further than in controls (Fig 4C). Contact probability features at > 10 Mb remain largely unchanged in both *Scc1*$^\Delta$ and *Wapl*$^\Delta$ $P_c(s)$ curves. Therefore, differences in long-range interactions (> 10 Mb) between maternal and paternal chromatin are cohesin-independent. Thus, we conclude that cohesin is directly involved in shaping the $P_c(s)$ curve up to ~1 Mb, and its effect is a deviation in contact probability above the $s^{-1.5}$ power law in mouse zygotic chromatin.

### Average extruded loop sizes can be derived from $P_c(s)$ curves and simulations

To help elucidate the mechanism of loop formation by cohesin, we developed a new method for analysis of $P_c(s)$ curves aiming to derive sizes of extruded loops and linear density of cohesin. We developed and tested this method using polymer simulations of loop extrusion, where sizes of loops and linear density of extruders are either set or can be directly measured. Our analysis shows that average loop sizes and cohesin density can be found by studying the

derivative of the $P_c(s)$ curve in log–log space, that is, the slope of $log(P_c(s))$ (Fig EV4A): The location of the maximum of the derivative curve (i.e., position of the smallest slope) closely matches the average length of extruded loops, and the depth of the local minimum at higher values of *s* increases with the linear density of loop-extruding cohesin in simulated chromatin (Fig EV4A). Note that sizes of extruded loops are smaller than the processivity of each cohesin, defined as the loop size extruded by unobstructed cohesin, suggesting some degree of crowding of cohesins on DNA (Appendix Fig S3), as expected theoretically (Fudenberg *et al*, 2016; Goloborodko *et al*, 2016) and illustrated schematically (Fig 1). We validate this approach using recent population Hi-C data for *Wapl*$^\Delta$ and control HAP1 cells (Haarhuis *et al*, 2017; Fig EV4B). We demonstrate that a twofold increase in cohesin density in *Wapl*$^\Delta$ can be inferred from the $P_c(s)$ curves, which matches experimentally measured values (Fig EV4A and B; see fig 4E in Haarhuis *et al*, 2017); moreover, we infer that the average size of an extruded cohesin loop in the HAP1 cells is ~120 kb in controls and ~300 kb in the *Wapl*$^\Delta$ condition.

We note that the extruded loops with the average size < 300 kb are different from peaks of Hi-C contact frequency, also referred to as "loops", that are typically formed by CTCF-rich TAD boundaries located up to 1 Mb from each other. Such peaks of interactions between boundaries also arise in simulations; they rarely represent a single boundary-to-boundary loop and are typically formed by a collection of much smaller cohesin-extruded loops that have bumped into each other and have stopped at TAD boundaries (Fig 1A). Due to the stochastic nature of cohesin loading and extrusion, the location of individual extruded loops formed by stalled cohesin varies from cell to cell and is not visible as an enrichment in Hi-C maps (Fig 1B). These loops, however, bring two boundaries closer to each other, and since boundary locations are set genomically, enrichment on interactions between boundaries becomes visible as peaks in Hi-C maps (here referred to as "Hi-C loops"). In all, this new method for analysis of $P_c(s)$ curves provides a framework for the interpretation of genome-wide contact probability and is complementary to identification of contact frequency peaks ("Hi-C loops") visible in Hi-C maps.

### Loop extrusion leads to differences in compaction of maternal and paternal chromatin

Interpreting our zygote data using the $P_c(s)$ curve analysis, we estimated that loop extrusion by cohesin results in an average extruded loop size of 60–70 kb in control G1 zygotes (Fig 4A). In contrast, in *Wapl*$^\Delta$ zygotes, the length of loops extruded by cohesin was doubled to 120 kb, whereas no loops could be detected in Scc1$^\Delta$ zygote data (Fig 4B and C). As a complementary approach, we performed polymer simulations at a range of cohesin density and processivity parameters and found values that provide the best agreement between simulations and experimental data, as measured by agreement of the $P_c(s)$ curves (Fig 4D–F): We obtain 74 kb as the average size of extruded loops for control zygotes (both maternal and paternal), 111 kb for paternal *Wapl*$^\Delta$ zygotic chromatin, and 165 kb for maternal *Wapl*$^\Delta$ zygotic chromatin. In addition, the best-matching models provide estimates for the processivity and linear density of cohesin in these cells: For control zygotes, we obtain a processivity

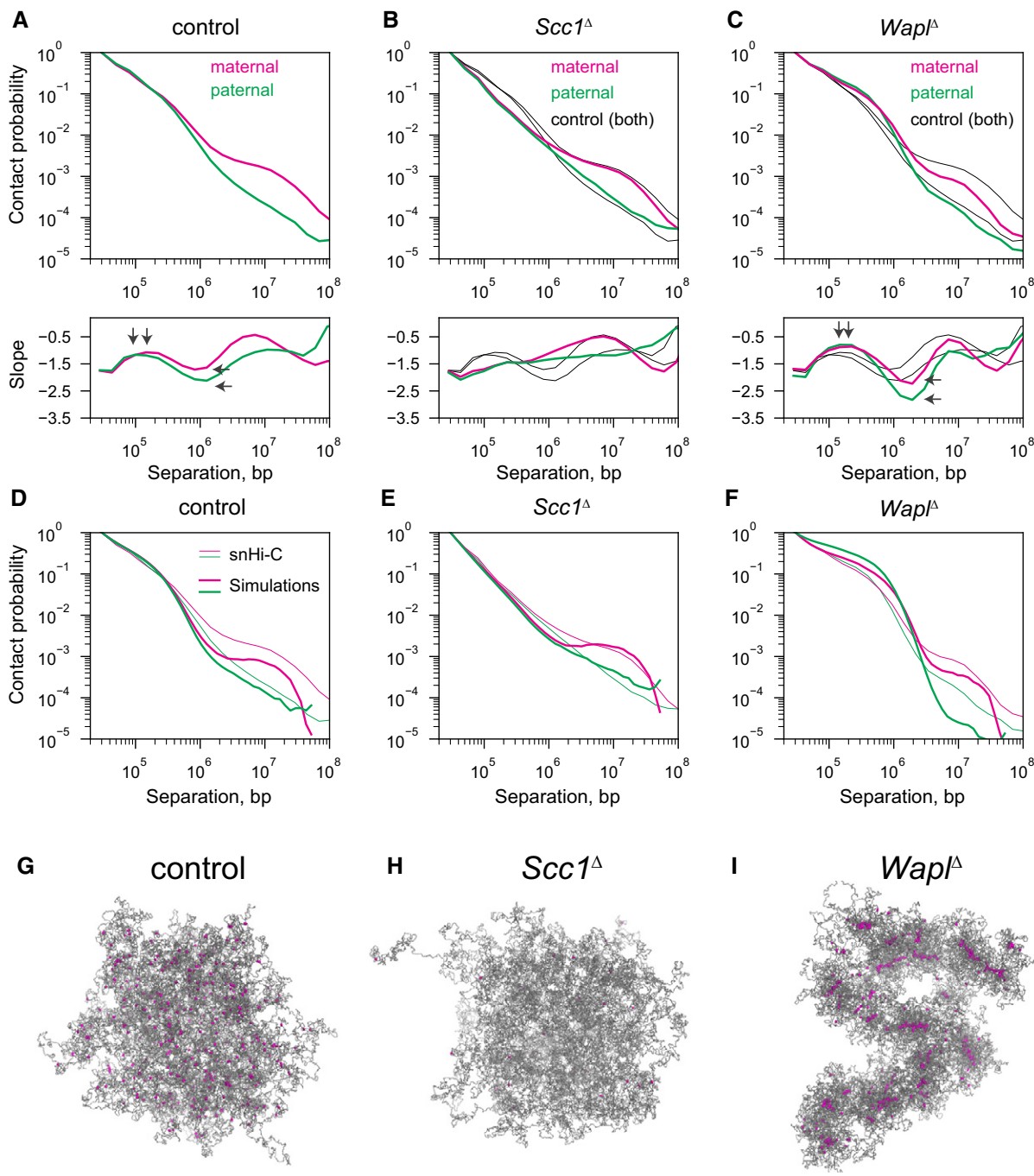

**Figure 4.  Differences in genome-wide contact probability, P$_c$(s), for chromatin loci separated by genomic distances, s, between conditions.**

A–C  Experimental P$_c$(s) for maternal and paternal chromatin for *Scc1* control, *Scc1*$^\Delta$, and *Wapl*$^\Delta$ conditions. Black solid lines in (B and C) show the control curves as a reference to guide the eye. Slopes of the log(P$_c$(s)) curves for each condition are shown in the subpanel below each P$_c$(s) plot. Vertical arrows on the slope subpanels indicate the maximum slope, which is used to infer the average size of cohesin-extruded loops; this analysis indicates that the average extruded loop size is approximately 60–70 kb in control zygotes and increases in the *Wapl*$^\Delta$ condition to over 120 kb. Horizontal arrows on the slope panels indicate the minimum slope, which can indicate cohesin linear density on the chromatin; notably, neither maternal nor paternal *Scc1*$^\Delta$ zygotes have a minimum slope suggesting very low cohesin density, whereas minima exist in both control and *Wapl*$^\Delta$ conditions. Data are based on n(*Wapl*$^{fl}$, maternal) = 7, n(*Wapl*$^{fl}$, paternal) = 6, n(*Wapl*$^\Delta$, maternal) = 8, n(*Wapl*$^\Delta$, paternal) = 7, n(Scc1$^{fl}$, maternal) = 13, n(*Scc1*$^{fl}$, paternal) = 17, n(*Scc1*$^\Delta$, maternal) = 28, and n(*Scc1*$^\Delta$, paternal) = 17 nuclei, from at least two independent experiments using two to three females per genotype each.

D–F  Simulated chromatin P$_c$(s) for the control, *Scc1*$^\Delta$, and *Wapl*$^\Delta$ conditions. Simulation P$_c$(s) curves shown in thick lines and experimental P$_c$(s) curves in thin lines.

G–I  Representative images of the simulated paternal chromatin fiber used for the P$_c$(s) calculations in panels (D–F). The chromatin fiber is colored in gray, and the locations of the cohesins are colored in purple.

Source data are available online for this figure.

of 120 kb and density of one cohesin per 120 kb (assuming one cohesin per loop extrusion complex). For *Wapl*$^\Delta$ zygotes, we require a much higher processivity of 480 kb in both maternal and paternal zygotes and a linear density of one cohesin per 120 kb in maternal and 60 kb in paternal chromosomes. We conclude that Wapl is mostly regulating cohesin processivity, as changes in linear density may be limited by the available number of cohesin complexes per nucleus.

To examine how Hi-C loops differ between the Wapl maternal and paternal genomes, we quantified their strength (Appendix Fig S1C) as done in Figs 2 and 3. We found that Hi-C loop strengths generally increased in the case of both maternal and paternal genomes. Analyzing the insulation in *Wapl*$^\Delta$ zygotes (see Materials and Methods) also showed stronger insulation at TAD/loop borders in paternal chromatin (Fig EV3D). The stronger Hi-C loops, stronger insulation, and higher cohesin density may all result from higher cohesin loading rate and reflect the transcriptionally permissive state specific for paternal chromatin (Adenot *et al*, 1997), suggesting that higher transcription leads to loading of additional cohesins, whose effects are exacerbated in *Wapl*$^\Delta$ where cohesin unloading is suppressed. This also suggests that transcription is not required for loop extrusion *per se*, as the maternal genome is thought to be transcriptionally inactive.

Next, we used microscopy to test whether these differences in loops between maternal and paternal chromatin lead to changes in chromatin compaction in *Wapl*$^\Delta$ zygotes. To monitor chromatin compaction, we expressed Scc1-EGFP in *Wapl*$^{\Delta/\Delta}$ and *Wapl*$^{fl/fl}$ oocytes, performed *in vitro* fertilization, and imaged zygotes by time-lapse microscopy (Appendix Fig S4A). Chromatin in *Wapl*$^\Delta$ zygotes is expected to form "vermicelli", prophase-like chromosomes with cohesin-enriched axial structures that can be detected by visualization of Scc1 (Lopez-Serra *et al*, 2013; Tedeschi *et al*, 2013). Scc1-EGFP formed a uniform diffuse pattern in the nuclei of control zygotes (Appendix Fig S4B). In contrast, Scc1-EGFP showed a non-homogeneous distribution in maternal and paternal nuclei of *Wapl*$^\Delta$ zygotes (Appendix Fig S4C). This distribution might reflect vermicelli that are obscured due to the presence of endogenous Scc1 within cohesin complexes, leading to a high background of free Scc1-EGFP. To ensure that all cohesin contains Scc1-EGFP, we expressed Scc1-EGFP in *Scc1*$^{\Delta/\Delta}$*Wapl*$^{\Delta/\Delta}$ oocytes (Fig 5A–C; Movies EV1 and EV2; Appendix Fig S5A and B). Indeed, this approach increased the detection of vermicelli as worm-like structures in both nuclei of *Scc1*$^\Delta$*Wapl*$^\Delta$ zygotes (Fig 5B and C; Movie EV2; Appendix Fig S5C). Vermicelli-like structures were especially evident in maternal nuclei in both *Wapl*$^\Delta$ and

*Scc1*$^\Delta$*Wapl*$^\Delta$ zygotes. Vermicelli formation occurs prior to the major ZGA (Aoki *et al*, 1997; Hamatani *et al*, 2004), consistent with the idea that transcription is not essential for Hi-C loop formation (Du *et al*, 2017; Ke *et al*, 2017). We conclude that inactivation of cohesin release leads to vermicelli formation in maternal and paternal zygotic chromatin.

To quantify maternal and paternal chromatin compaction, we examined DNA morphology at higher resolution in fixed zygotes. Both maternal chromatin and paternal chromatin are compacted into vermicelli-like structures and are revealed most clearly in individual z-sections of *Wapl*$^\Delta$ zygotes (Fig 6A and B). We observed a significant change in the coefficient of variation in intensity between control and *Wapl*$^\Delta$ zygotes (Fig 6C; Appendix Fig S6; *P*-value = $1.88 \times 10^{-7}$). Additional DAPI-intense structures surrounding the prenucleolar regions were visible specifically in maximum-intensity projections in the maternal nucleus (*n* = 25/33 zygotes; Fig 6A and B), indicating a higher degree of compaction in maternal than paternal chromatin. These DAPI-intense structures likely correspond to the more prominent vermicelli observed in maternal nuclei in time-lapse movies (Fig 5B and C; Appendix Fig S4C; Movie EV2). Quantification of the texture in images using the gray-level co-occurrence matrices revealed that the contrast between pixels is stronger in maternal than paternal nuclei (Fig 6D and Appendix Figs S7 and S8), implying a more structured and less homogeneous nuclear architecture. To study the DAPI-intense structures, we performed additional segmentation analysis and compared the size distributions of identified objects between conditions and nuclei. The size of DAPI-intense structures significantly increased in *Wapl*$^\Delta$ zygotes (*P*-values: $1.25 \times 10^{-11}$ and $8.23 \times 10^{-28}$ for maternal and paternal nuclei, respectively; Fig 6E). Maternal nuclei contain slightly bigger objects than paternal nuclei (*P*-value: 0.00014), which might reflect stronger vermicelli. We suggest that inactivating cohesin release has a differential effect on chromatin compaction of maternal and paternal chromatin.

To corroborate the major reorganization observed by microscopy and snHi-C in *Wapl*$^\Delta$ zygotes, we examined our polymer simulations of *Wapl*$^\Delta$ conditions to see whether the 3D organization of cohesins in modeled conformations displayed preferentially "axially enriched" structures (Appendix Fig S3). We found consistently that vermicelli are visible in the paternal *Wapl*$^\Delta$ chromatin simulation, but are not visible in controls (Fig 4G and I); at odds with expectations, maternal chromatin formed weaker vermicelli (Fig EV4C; Appendix Fig S3). This result suggests that some other processes beyond loop extrusion may contribute to formation of vermicelli in

---

**Figure 5.   Live-cell imaging of vermicelli formation in wild-type and *Scc1*$^\Delta$*Wapl*$^\Delta$ zygotes expressing Scc1-EGFP and H2B-mCherry.**

A   Germinal vesicle-stage oocytes were injected with mRNA encoding H2B-mCherry to mark chromosomes (magenta) and Scc1-EGFP to label cohesin (green), matured to meiosis II, fertilized *in vitro*, and followed by time-lapse microscopy.

B   Still images of live wild-type zygotes expressing Scc1-EGFP and H2B-mCherry (*n* = 4 zygotes, from one experiment using two females). Top row: z-stack maximum-intensity projection of zygotes. Middle and bottom row: z-slices of the cropped areas (top left) showing paternal and maternal nuclei separately. Images were adjusted in brightness/contrast in individual imaging channels in the same manner for z-stacks and for the single z-slices. Scale bars: 10 μm. Hours after start of IVF are given.

C   Still images of live *Scc1*$^\Delta$*Wapl*$^\Delta$ zygotes expressing Scc1-EGFP and H2B-mCherry (*n* = 3 zygotes, from one experiment using two females). Top row: z-stack maximum-intensity projection of zygotes. Middle and bottom row: z-slices of the cropped areas (top left) showing paternal and maternal nuclei separately. Arrows indicate Scc1-EGFP-enriched structures. Images were adjusted in brightness/contrast in individual imaging channels in the same manner for z-stacks and for the single z-slices. Scale bars: 10 μm. Hours after start of IVF are given.

Data information: Experiments shown in (B and C) were performed individually, but under the same conditions using the same mRNA injection mix.

**A**

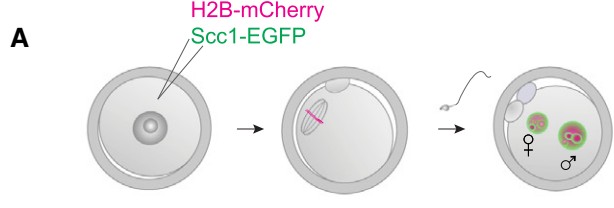

**B**   wild-type

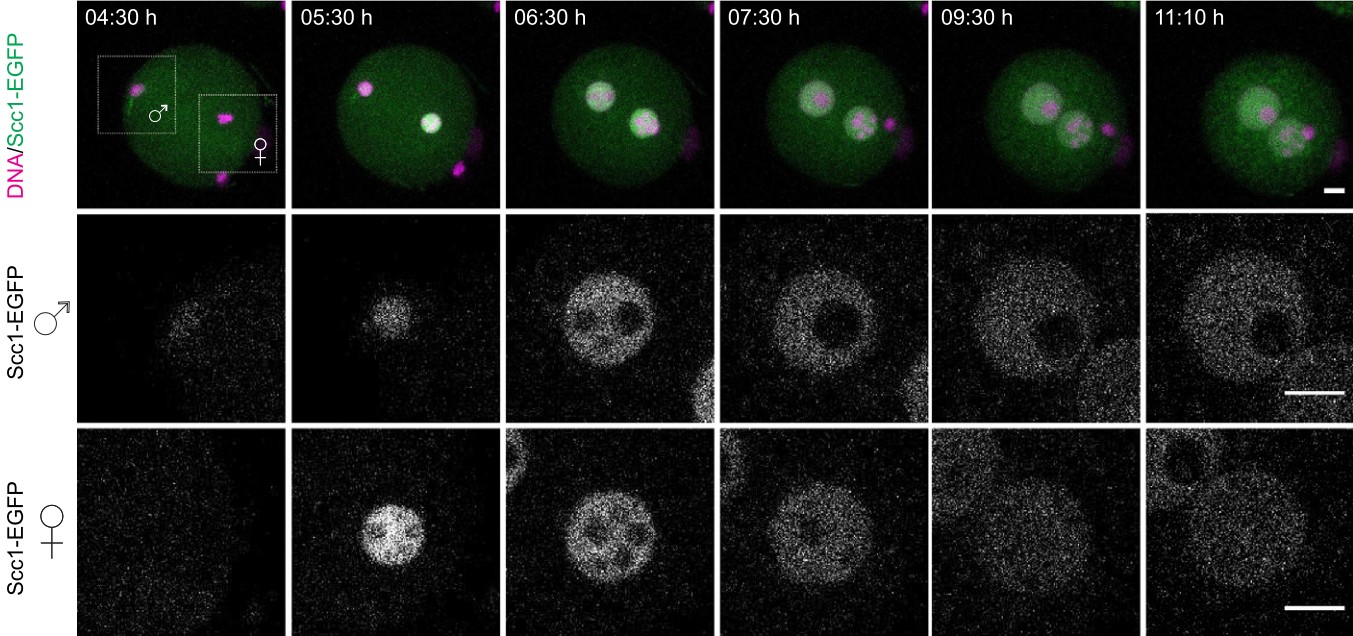

**C**   *Scc1ᐞWaplᐞ*

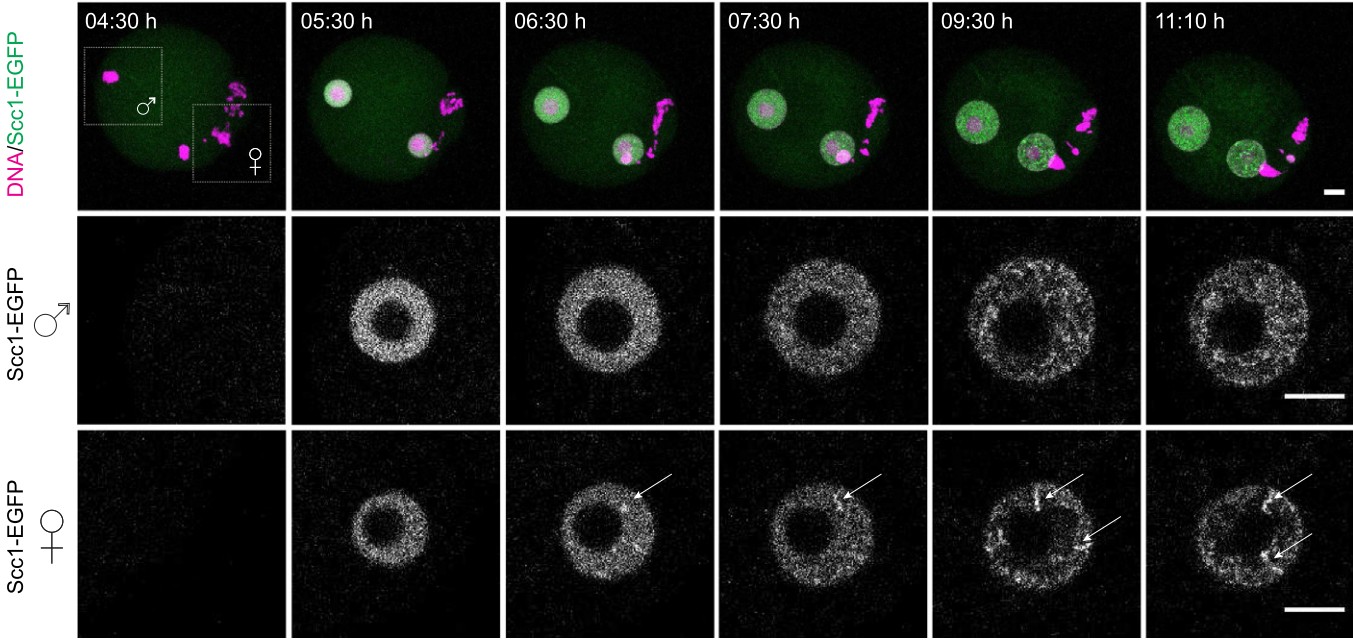

**Figure 5.**

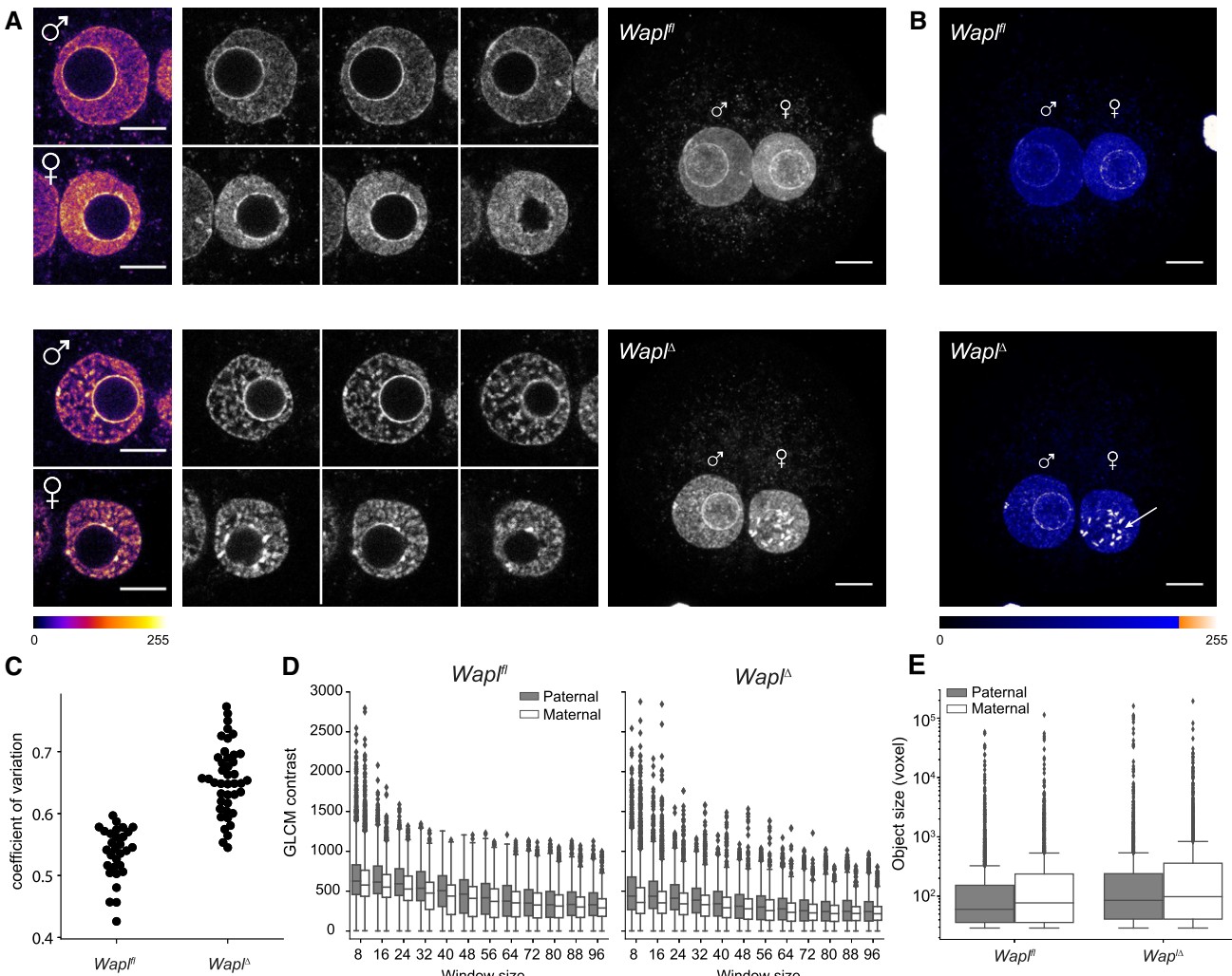

**Figure 6.  Distinct maternal and paternal chromatin compaction in *Wapl*$^\Delta$ zygotes.**

A   Representative images of paternal and maternal nuclei stained with DAPI of *Wapl*$^{fl}$ (*n* = 15) and *Wapl*$^\Delta$ (*n* = 33) zygotes (from two independent experiments using two females per genotype; see also Appendix Fig S5). Top: *Wapl*$^{fl}$; bottom: *Wapl*$^\Delta$. Left: cropped z-slices from the middle section of the nucleus in fire lookup table (Image J). Middle: cropped z-slices of nuclei separated by 3 μm. Right: maximum-intensity projection (MIP) of zygotes. Settings were adjusted for z-slices and MIP individually but in the same manner for *Wapl*$^{fl}$ and *Wapl*$^\Delta$ zygotes. Images were adjusted in brightness/contrast in the individual imaging channels using ImageJ. Scale bars: 10 μm.

B   MIP of zygotes seen in (A) with blue ramp lookup table to visualize difference in maternal and paternal vermicelli formation around prenucleolar bodies. Arrow indicates additional DAPI-intense structures in maternal zygotic nuclei. Images were adjusted in brightness/contrast in the individual imaging channels using ImageJ. Scale bars: 10 μm.

C   Coefficient of variation of DAPI intensity for nuclei of *Wapl*$^{fl}$ (*n* = 15) and *Wapl*$^\Delta$ (*n* = 21) zygotes (P-value = 1.88 × 10$^{-7}$, Mann–Whitney U-test).

D   Boxplots showing gray-level co-occurrence matrix (GLCM) contrast (local variation of intensity) in paternal (gray) and maternal (white) nuclei in *Wapl*$^{fl}$ (*n* = 15) and *Wapl*$^\Delta$ (*n* = 13) zygotes with increasing window sizes. Horizontal lines of the boxplots represent the medians, box limits show the first and third quartiles, whiskers extend by 1.5 * interquartile range from the limits of the box. Two outliers (maternal *Wapl*$^\Delta$ window 8) with values 3,242.7 and 4,037.4 are not shown.

E   Boxplots showing size of detected bright objects (voxels) inside paternal (gray) and maternal (white) nuclei in *Wapl*$^{fl}$ (*n* = 15) and *Wapl*$^\Delta$ (*n* = 21) zygotes; note the log scale on *y*-axis.

Source data are available online for this figure.

maternal zygotes. Nevertheless, both our snHi-C data and microscopy show that loop formation differs for zygotic maternal and paternal genomes when cohesin release is prevented by Wapl depletion. By regulating cohesin release, Wapl thus maintains interphase chromatin in a less compact state; moreover, it restricts the size of extruded cohesin loops and density of chromatin-associated cohesin.

**Cohesin loop extrusion limits inter-chromosomal interactions**

Population and single-cell Hi-C studies have revealed that interactions between non-sister chromatids (*trans*-contacts) are diminished during mitosis (Naumova *et al*, 2013; Nagano *et al*, 2017). A possible interpretation is that a more compact, linearly ordered chromosome directly affects the frequency of inter-chromosomal

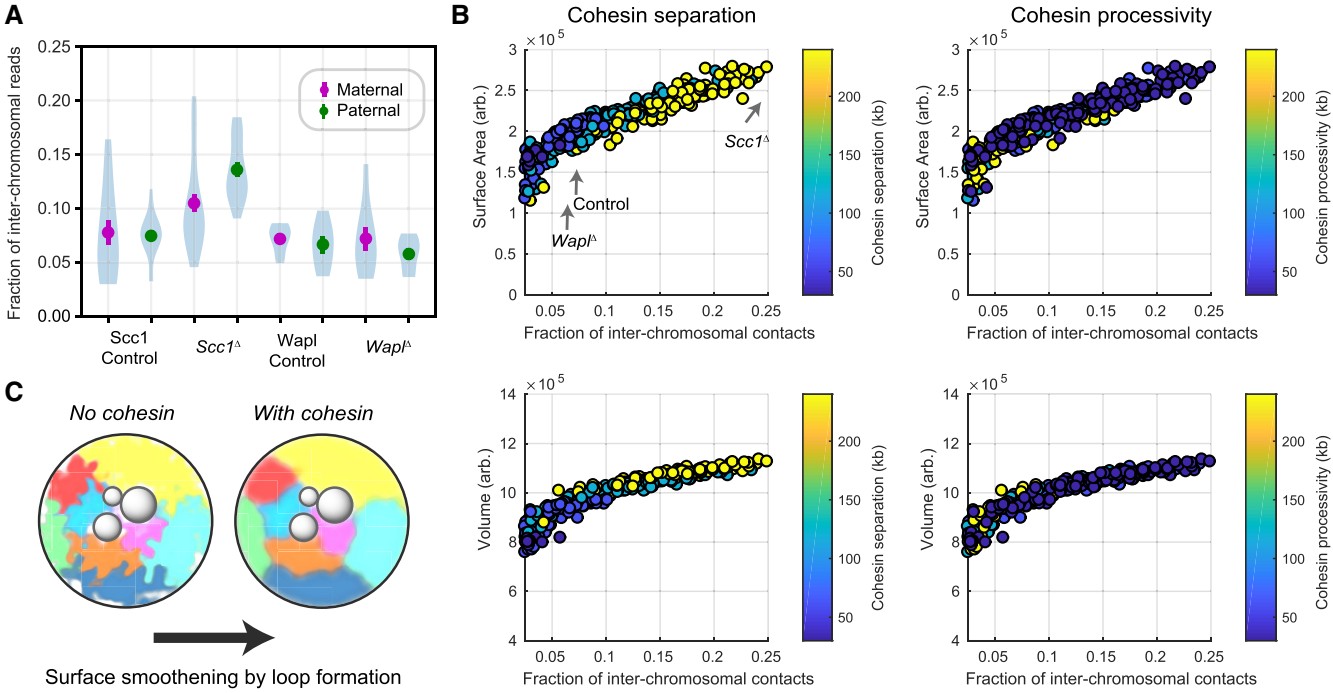

**Figure 7.  Influence of cohesin on inter-chromosomal contacts.**

A  The number of snHi-C contacts mapping to regions on distinct chromosomes as a fraction of the total number of mapped contacts is shown for each of the experimental conditions. Error bars are the standard error of the mean. The distribution of values from individual nuclei is shown in blue as a violin plot; the maximum extent of the density distributions reflects the range of the individual data points ($n(Wapl^{fl}$, maternal) = 7, $n(Wapl^{fl}$, paternal) = 6, $n(Wapl^{\Delta}$, maternal) = 8, $n(Wapl^{\Delta}$, paternal) = 7, $n(Scc1^{fl}$, maternal) = 13, $n(Scc1^{fl}$, paternal) = 17, $n(Scc1^{\Delta}$) = 28, and $n(Scc1^{\Delta}$) = 17 nuclei; data are based on at least two independent experiments using 2–3 females per genotype each).

B  Spatial, geometric properties of simulated chromatin undergoing loop extrusion for different loop extrusion parameters. The fraction of inter-chromosomal contacts was calculated using a Hi-C cutoff radius of 5 monomers (75 nm). The surface area and volume of the simulated chromatin fiber were calculated from the concave hull, and an effective radius for each monomer equal to the Hi-C cutoff radius was used (see Materials and Methods).

C  A schematic model illustrating that cohesin loop extrusion can modulate the surface area smoothness of chromosomes and reduce the frequency of inter-chromosomal interactions.

Source data are available online for this figure.

interactions. To investigate whether vermicelli chromosomes are more mitotic-like, and to test whether cohesin might play a role in chromosome compaction, we quantified the levels of *trans*-contacts, in zygotic chromatin by snHi-C (Fig 7A; see Materials and Methods; Tables EV1 and EV2). We found inter-chromosomal contact frequencies of 8% for nuclei in interphase (G1/S or G2), consistent with values reported for mouse ES cells at a similar cell cycle stage (Nagano *et al*, 2017). Interestingly, $Wapl^{\Delta}$ zygotes had reduced *trans* interaction fractions, with a mean value of 6% for paternal zygotic chromatin that is closer to values reported for early G1 (Nagano *et al*, 2017) but not significantly different from controls ($P < 0.2$, Mann–Whitney $U$-test). In contrast, $Scc1^{\Delta}$ cells showed significantly larger *trans* interaction fractions as compared to controls (Fig 7A; an over 40% increase, $P < 0.02$, Mann–Whitney $U$-test). These results suggest a possible novel role for chromosomal Scc1-cohesin in reducing interaction frequencies between non-sister chromatids.

To investigate the mechanism by which cohesin modulates inter-chromosomal interactions, we turned to polymer simulations of loop extrusion. We tested how varying cohesin processivity and linear density affected absolute numbers of contacts within and between chromosomes (Appendix Fig S9A). We found that an increase in processivity or density of cohesins resulted in an increase in intra-chromosomal contacts and a decrease in the absolute and relative *trans*-chromosomal contacts (Appendix Fig S9A). Thus, simulations suggest that cohesin can regulate frequencies of contacts between chromosomes.

To better understand how loop extrusion that operates at the sub-megabase scale can affect inter-chromosomal contacts, we examined the effects of loop extrusion on the sizes of chromosomes and shapes of their surfaces (Figs 7B and 8A). We varied cohesin processivity and linear density and measured their effects on the simulated chromatin volume and surface area defined from the polygon that covers the modeled chromosomes (concave hull; see Materials and Methods). We found that an increase in processivity and linear density of cohesins from $Scc1^{\Delta}$ to control to $Wapl^{\Delta}$ levels led to a gradual decrease in the number of *trans* interactions, a decrease in volume, and a decrease in surface area (Figs 7B and 8B); this trend was not sensitive to the choice of simulated Hi-C capture radius, or chromosome monomer radius for the convex hull measurement (Appendix Fig S9B and C). Interestingly, we found that the chromosome surface area was a good predictor of the fraction of inter-chromosomal interactions changing over 80% almost linearly from the simulated $Wapl^{\Delta}$ to $Scc1^{\Delta}$ conditions; however,

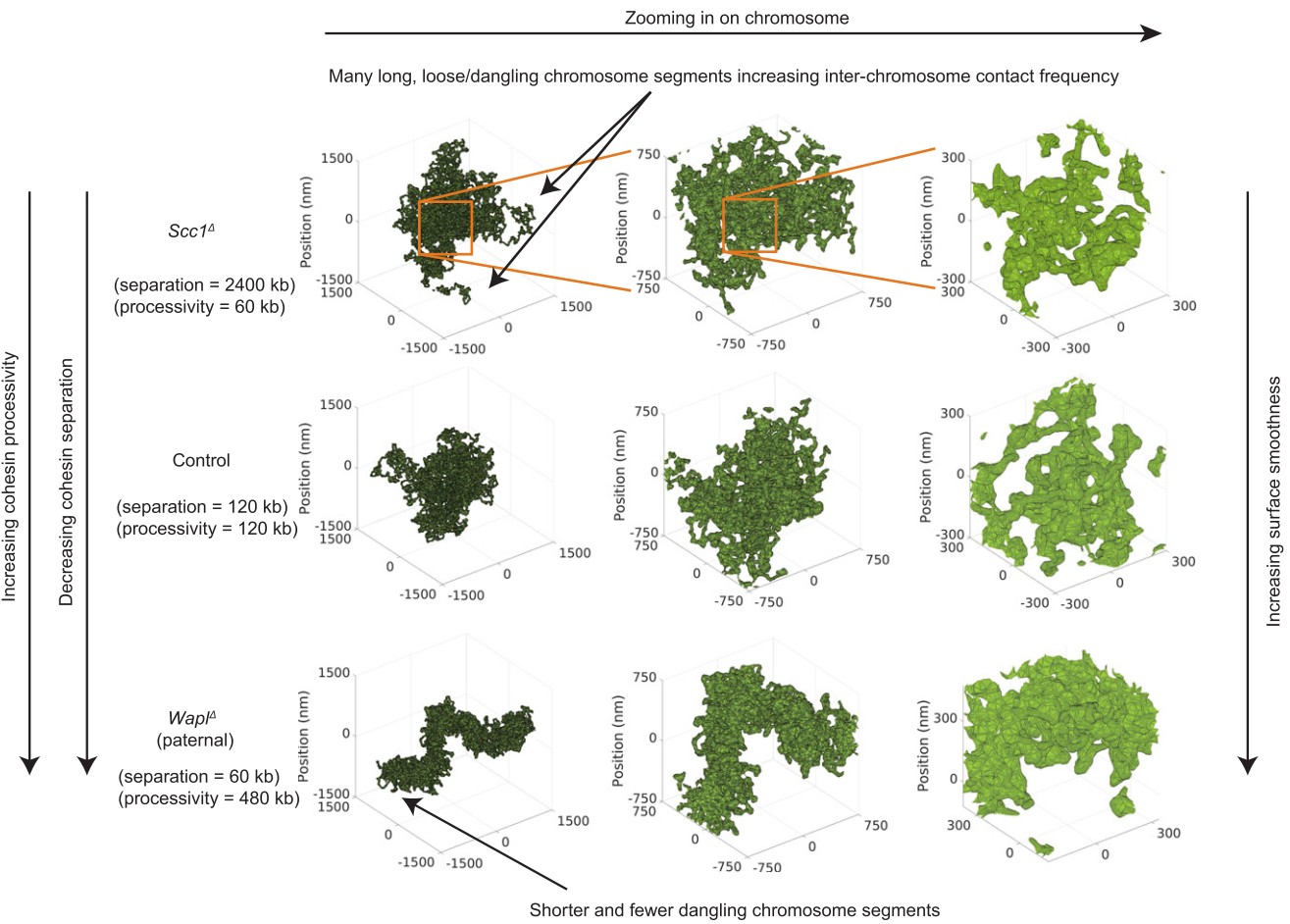

**Figure 8.  Effect of loop extrusion on the simulated chromatin surface area, volume, and inter-chromosomal interactions.**

Representative polymer conformations of simulated chromatin undergoing loop extrusion. The rendered surface is the alpha shape (concave hull polygon) created using spheres centered on chromosome monomers. The monomers have radius 75 nm, which were chosen to be equal to the simulated Hi-C capture frequency. With increasing cohesin density and processivity, the chromosome compacts and becomes more linearly ordered and the concave hull surface becomes "smoother".

whereas volume was predictive, it changed by only 40% and was nonlinear (Fig 7B; Appendix Fig S9B).

By visualizing polymer conformations for low and high cohesin densities, we found that a decrease in the number of extruded loops led to a surface roughening, whereas increased compaction by loop extrusion smoothened out the polymer surface, resulting in fewer inter-chromosomal contacts (Figs 7C and 8A). These simulations demonstrate that loop extrusion operating at < 1-Mb scale can affect long-range interactions by modulating the surface area of chromosomes, leading to changes in inter-chromosomal interaction frequencies. Super-resolution microscopy of continuously stained chromosomal regions may be able to observe the predicted roughening of chromosomal surfaces upon loss of cohesin.

## Discussion

Our data support a direct role of cohesin in the formation or maintenance of chromatin loops and TADs. Cohesin was identified over two decades ago for its role in chromosome segregation, sister

chromatid cohesion, and DNA damage repair (Peters *et al*, 2008). More recent studies have shown that cohesin colocalizes with CTCF and is associated with TADs and chromatin loops (Wendt *et al*, 2008; Dixon *et al*, 2012; Nora *et al*, 2012; Phillips-Cremins *et al*, 2013; Rao *et al*, 2014; Hansen *et al*, 2017; Nora *et al*, 2017), which implicated cohesin as a regulator of intra-chromosomal structure. Since chromatin loops and TADs may have functional roles in gene regulation, such as preventing aberrant expression of genes (Flavahan *et al*, 2015; Lupiáñez *et al*, 2015; Franke *et al*, 2016), it has become a major endeavor to understand to what degree cohesin is involved in shaping chromatin structure. Early studies directly degrading or knocking out cohesin showed only mild effects on chromatin structure (Seitan *et al*, 2013; Sofueva *et al*, 2013; Zuin *et al*, 2014).

We show that genetic deletion of the Scc1 subunit of cohesin in mouse oocytes abolishes formation or maintenance of loops and TADs in the one-cell embryo. In contrast, chromatin loops are larger on average when cohesin release from chromosomes is prevented by Wapl depletion. Together, these results demonstrate that cohesin is essential for loops and TADs, and show that cohesin directly

regulates their structure, consistent with recent studies that were published while this paper was under review. A recent study in a human cancer cell line (Rao *et al*, 2017) shows loss of loops and TADs upon acute degradation of Scc1/Rad21; similar results using this approach in HeLa cells were obtained in a recent preprint (Wutz *et al*, 2017). Another study of a HAP1 human cell line (Haarhuis *et al*, 2017) demonstrates that *Wapl* deletion leads to higher density of cohesin on DNA and increases contact frequency of distant Hi-C loops. Finally, a recently published study achieved depletion of chromatin-associated cohesin by deletion of *Nipbl* in post-mitotic liver cells, which led to disappearance of loops and TADs (Schwarzer *et al*, 2017).

We extend these studies by uniquely obtaining both a decrease and an increase in cohesin, relative to the wild type, in the same biological system. This allowed us to gain insights into the fundamental principles of chromatin organization, developing a single polymer model that was able to reproduce chromosomal phenotypes of the three tested conditions, providing quantitative estimates of characteristics of cohesin-mediated loop extrusion, and making predictions about the effect of loop extrusion on long-range interactions by roughening of chromosomal surfaces. Our work also diverges from these reports in that we show cohesin is essential for forming loops and TADs starting from the one-cell embryo, which was hitherto unclear.

Crucially, our system enabled us to study how cohesin differentially affects the establishment of higher-order structure in maternal and paternal genomes that undergo reprogramming to totipotency. Interestingly, differences in maternal and paternal chromatin loops became more evident in *Wapl*$^{\Delta}$ zygotes. As in controls, paternal chromatin loops were stronger and TADs were more insulating than in maternal chromatin. Unlike controls, loop sizes differed by an estimated 60 kb, with longer loops present in the maternal genome. By microscopy, we also observed differences in global chromatin compaction between maternal and paternal genomes in *Wapl*$^{\Delta}$ zygotes. We speculate that the differences are due to a combination of distinct epigenetic modifications and loop extrusion dynamics.

Our data strongly support a model that cohesin forms loops and TADs by the mechanism of active loop extrusion (Sanborn *et al*, 2015; Fudenberg *et al*, 2016), and provide a quantitative rationale for the longer loop lengths in the *Wapl*$^{\Delta}$ zygotes. Our polymer simulations suggest that the key determinants for global genome organization by cohesins are their density and processivity, which is the product of residence time and extrusion velocity. Longer chromatin loop sizes in *Wapl*$^{\Delta}$ zygotes are quantitatively consistent with an about fourfold increase in cohesin processivity in the absence of Wapl, which results in an about 50% increase in the sizes of extruded loops as estimated from the derivative of log ($P_c(s)$). Our present data do not distinguish whether increase in processivity reflects an increase in loop-extruding speed, residence time, or both, but this is an interesting avenue for future research. Interestingly, sizes of extruded loops are smaller than the processivity since extrusion is obstructed by interactions of boundary elements (with CTCF among them) and other chromatin-associated cohesins. In support of the model of active loop extrusion, Wang and coworkers recently provided the first direct *in vivo* evidence that condensins, which are related to cohesins, actively translocate on bacterial chromatin and align flanking chromosomal DNA (Tran

*et al*, 2017; Wang *et al*, 2017). A recent *in vitro* study has since demonstrated that eukaryotic yeast condensins are mechanochemical motors that translocate along DNA in an ATP-dependent fashion (Terakawa *et al*, 2017). Thus, it is likely that eukaryotic cohesins employ active loop extrusion to form chromatin loops and TADs, but we cannot rule out the possibility that accessory factors aid the extrusion process.

In contrast to our findings, two recent reports of the higher-order chromatin organization in mammalian embryos suggested that the mammalian zygote genome is largely unstructured (Du *et al*, 2017; Ke *et al*, 2017). In both studies, no or obscure TADs were detected in embryos before the eight-cell stage (Du *et al*, 2017; Ke *et al*, 2017), where TADs were detected using insulation scores and directionality index analysis (Dixon *et al*, 2012; Giorgetti *et al*, 2016) with a large window size (0.5–1 Mb). We note that nonzero insulation scores or directionality indices do not necessarily reflect the existence of a TAD since these metrics cannot distinguish TADs from compartments without other information; weak compartments in zygotes can affect insulation scores or directionality indices. To further investigate whether TADs and loops exist in zygotes, we re-analyzed data from these studies. Using known positions of TADs and loops, we identified TADs and loops at all embryonic development stages. To exclude biases introduced by TAD positions used in the analysis, we tested TADs identified in many diverse cell types as well as TADs called *de novo* in bulk Hi-C of inner cell mass cells of blastocyst embryos (Du *et al*, 2017). Our ability to detect TADs in re-analyses of bulk Hi-C studies (Du *et al*, 2017; Ke *et al*, 2017) can be attributed to the higher statistical power of methods that we employed: Not only did we aggregate TADs from positions called in population Hi-C data, but we also used observed-over-expected maps to correct for $P_c(s)$ specific for the used Hi-C map and rescaled TADs of different sizes (100 kb–1 Mb), allowing to depict the structure of the TAD body independently of TAD sizes upon averaging. The lack of rescaling of TADs (as well as different normalization) in the original analysis could have led to blurring of signal in aggregate analysis. We further validated our method by visual inspection of Hi-C maps that showed both regions lacking contact enrichment and other regions containing domain structures. We furthermore show that the structures detected by aggregate analysis depend critically on cohesin, which is in line with its proposed role in loop and TAD formation.

Loops and TADs are weaker in zygotes than for later stage embryos, consistent with previous reports (Du *et al*, 2017; Ke *et al*, 2017). There are several possible explanations for this phenomenon, such as weaker or fewer boundary elements, lower rate of cohesin loading, or lower cohesin processivity. The difference in processivity is unlikely as our analysis suggests a similar processivity in paternal zygotic chromatin and HAP1 cells. On the other hand, we show both TAD and loop strengths are visually greater in the early G1 paternal zygotic genome, but these differences disappear in G2 as both genomes approach the major ZGA. We thus suggest that the weaker structural features seen in the zygotic genome arise due to either paucity of boundary elements for cohesin loop extrusion or lower amounts of chromatin-associated cohesin.

Unexpectedly, we discovered that cohesin-dependent chromosome compaction reduces inter-chromosomal interactions in interphase. We therefore propose a model in which the surface

roughness of chromosomes affects inter-chromosomal interactions and absence of cohesin leads to more interdigitation between chromosomes. We speculate as to what might be the functional consequences of increased inter-chromosomal interactions due to interphase chromosome decompaction. Given that topoisomerases cannot distinguish between DNA strands in *cis* and in *trans*, it is conceivable that increased number of *trans* interactions could lead to catenations that can be damaging during chromosome segregation. We therefore propose that the ancestral role of cohesin in forming intra-chromosomal loops during interphase could help promote proper chromosome segregation during cell division.

Our model of cohesin as a chromatin surface area regulator also raises important new points. If the active formation of loops can reduce inter-chromosomal interactions, then it is conceivable that loop formation creates local neighborhoods on the chromatin fiber that also reduce the frequency of interactions with more distal segments of chromatin on the same chromosome. We speculate that the formation of loops can have important implications for reducing spurious enhancer–promoter looping interactions by reducing interdigitation between distant regions of the same chromosome.

In all, our work establishes which higher-order chromatin structures are built shortly after fertilization in the mammalian zygote. The differences in maternal and paternal loops generated by cohesin-dependent loop extrusion provide an entry point to understanding how the two genomes change from a transcriptionally silent and terminally differentiated state to a transcriptionally active and totipotent embryonic state.

# Materials and Methods

### Mice

The care and use of the mice were carried out in agreement with the authorizing committee according to the Austrian Animal Welfare law and the guidelines of the International Guiding Principles for Biomedical Research Involving Animals (CIOMS, the Council for International Organizations of Medical Sciences). Mice were kept at a daily cycle of 14-h light and 10-h dark with access to food *ad libitum*. All mice were bred in the IMBA animal facility. *Scc1*$^{fl/fl}$ mice were bred on a mixed background (B6, 129, Sv). *Wapl*$^{fl/fl}$ mice were bred on a primarily C57BL/6J background. *Scc1*$^{fl/fl}$ *Wapl*$^{fl/fl}$ mice were bred on the same mixed background as *Scc1*$^{fl/fl}$ mice. Experimental mice were obtained by mating of homozygous floxed females to homozygous floxed males carrying *Tg(Zp3-Cre)* (Lewandoski *et al*, 1997; Lan *et al*, 2004). To obtain zygotes, B6CBAF1 stud males were mated to *Scc1*$^{fl/fl}$ *Tg(Zp3-Cre)*, while C57BL/6J stud males were used for mating *Wapl*$^{fl/fl}$ *Tg(Zp3-Cre)* females. Sperm for *in vitro* fertilization of *Scc1*$^{fl/fl}$ *Wapl*$^{fl/fl}$ *Tg(Zp3-Cre)* oocytes was obtained from B6CBAF1 stud males, sperm for *in vitro* fertilization of *Wapl*$^{fl/fl}$ *Tg(Zp3Cre)* oocytes was obtained from C57BL/6J stud males.

No statistical methods were used to estimate sample size. No randomization or blinding was used.

### Zygote collection

To obtain zygotes, 3- to 5-week-old female mice were superovulated by intraperitoneal injection of PMSG (pregnant mare's serum gonadotropin; 5 IU, Folligon, Intervet or 5 IU, Prospecbio) followed by hCG (human chorionic gonadotropin; 5 IU, Chorulon, Intervet) injection 48 h later. Females were mated to wild-type stud males overnight. The following morning, zygotes were released from the ampullae and treated with hyaluronidase to remove cumulus cells.

### Single-nucleus Hi-C

Single-nucleus Hi-C was carried out as described before (Flyamer *et al*, 2017). After pronuclear extraction, *Scc1*$^{fl/fl}$ *Tg(Zp3-Cre)* pronuclei used in the experiments were fixed around 19–22 h post-hCG injection (corresponding to about 7–10 h post-fertilization) and therefore are expected to be in G1/S phase of the cell cycle. *Wapl*$^{fl/fl}$ *Tg(Zp3-Cre)* were fixed later around 23–27.5 h post-hCG injection (corresponding to about 11–15.5 h post-fertilization) and are expected to be in S/G2 phase of the cell cycle. To obtain G2-phase data, zygotes were fixed 27 h post-hCG injection (corresponding to about 15 h post-fertilization) and lysed, and pronuclei were separated into different wells after SDS lysis according to their size. No blinding or randomization was used for handling of the cells.

Briefly, after pronuclei were isolated, they were fixed in 2% formaldehyde for 15 min and then lysed on ice in lysis buffer (10 mM Tris–HCl pH 8.0, 10 mM NaCl, 0.5% (v/v) NP-40 substitute (Sigma), 1% (v/v) Triton X-100 (Sigma), 1× Halt™ Protease Inhibitor Cocktail (Thermo Scientific)) for at least 15 min. The pronuclei were washed once through PBS and 1× NEB3 buffer (NEB) with 0.6% SDS, in which they were then incubated at 37°C for 2 h with shaking in humidified atmosphere. Pronuclei were washed once in 1× DpnII buffer (NEB) with 2× BSA (NEB), and then, chromatin was digested overnight in 9 μl of the same solution but with 5 U DpnII (NEB). The nuclei were then washed once through PBS, then through 1× T4 ligase buffer (50 mM Tris–HCl, 10 mM MgCl$_2$, 1 mM ATP, 10 mM DTT, pH 7.5). The nuclei were incubated in the same buffer but with 5U T4 DNA ligase (Thermo Scientific) at 16°C at 50 rpm for 4.5 h, and then for 30 min at room temperature. Whole-genome amplification was performed using illustra GenomiPhi v2 DNA amplification kit (GE Healthcare) with decrosslinking nuclei at 65°C overnight in sample buffer. High molecular weight DNA was purified using AMPure XP beads (Beckman Coulter), and 1 μg was used to prepare Illumina libraries for sequencing (by VBCF NGS Unit, csf.ac.at) after sonicating to ~300–1,300 bp. Libraries were sequenced on HiSeq 2500 v4 with 125-bp paired-end reads (at VBCF) or on NextSeq high-output lane with 75-bp paired-end reads (at Wellcome Trust Clinical Research Facility, Edinburgh), between 10 and 24 cells per lane.

### DNA and Scc1 staining

After zygote collection, the cells were fixed in 4% PFA for 30 min, before permeabilization in 0.2% Triton X-100/PBS (PBSTX) for 30 min. Cells were then blocked in 10% goat serum (Dako) in PBSTX either at 4°C overnight or for several hours at 4°C followed by incubation at room temperature. Cells were incubated overnight at 4°C in primary antibody (anti-Scc1, Millipore #05-908, 1:250). After washing in blocking solution for at least 30 min, incubation with the secondary antibody (anti-mouse IgG (H + L), Thermo Fisher Scientific #A-11001, 1:500) was carried out for 1 h at room

temperature. Another set of washing steps in 0.2% PBSTX was followed by a quick PBS wash and mounting of the cells in Vectashield containing DAPI (Vector Labs) using imaging spacers (Sigma-Aldrich). *In situ* fixed zygotes were imaged on a confocal microscope (LSM780, Zeiss, ZEN black) using a 63×, 1.4NA oil objective. The presence of DNA compaction reminiscent of vermicelli in Wapl zygotes was classified using ImageJ and 3D visualization by Imaris (8.1.2). Brightness and contrast of images presented were adjusted using ImageJ software. No blinding or randomization was used for handling of the cells. Samples were excluded from the analysis if cells were not fertilized or in the wrong cell cycle phase (PN stage).

### Antibodies

Anti-Rad21 (anti-Scc1, 1:250, Millipore, Cat# 05-908; RRID: AB_417383); and goat anti-mouse IgG (H + L) cross-adsorbed secondary antibody (1:500, Alexa Fluor 488, Thermo Fisher Scientific, Cat# A-11001; RRID: AB_2534069).

### Live-cell imaging of Scc1-EGFP

*In vitro* fertilization after *in vitro* maturation was performed as described before (Ladstätter & Tachibana-Konwalski, 2016). Oocytes from 2- to 5-month-old females were isolated by puncturing of ovaries with hypodermic needles in the presence of 0.2 mM IBMX, 20% FBS (Gibco), and 6 mg/ml fetuin (Sigma-Aldrich). After microinjection of oocytes with H2B-mCherry (187 ng/μl) and Scc1-EGFP (260 ng/μl), oocytes were cultured for 1–1.5 h and then released from IBMX inhibition by washing in M16. Following *in vitro* maturation in the incubator (low-oxygen conditions: 5% $CO_2$, 5% $O_2$, 90% $N_2$; 37°C), cells were scored for extrusion of the first polar body and MII eggs were *in vitro* fertilized 10.5–12 h post-release from IBMX inhibition. The sperm was obtained from the *cauda epididymis* and *vas deferens* of B6CBAF1 stud males and was capacitated in fertilization medium (Cook) in a tilted cell culture dish for at least 30 min. Motile sperm from the surface of the dish was used for *in vitro* fertilization of the *in vitro* matured eggs. After 3–3.5 h, zygotes were washed in M16 and imaged. Live-cell imaging of zygotes microinjected with fluorescent fusion proteins was performed on a confocal microscope (LSM 800, Zeiss; ZEN blue) equipped with an incubation chamber suited for live-cell imaging (5% $CO_2$, 37°C). Zygotes were kept in ~3 μl cleavage medium (Research Vitro Cleave; Cooks Austria GmbH) under mineral oil (Sigma-Aldrich or Millipore) for the duration of the imaging. Movies were taken using a 63×, 1.20NA water immersion objective, taking 25 z-slices (48 μm) every 10 min. Brightness and contrast of images presented were adjusted using ImageJ software. No blinding or randomization was used for handling of the cells. Samples were excluded from the analysis if cells were not fertilized.

### snHi-C data analysis

snHi-C data were processed similarly as in Flyamer *et al* (2017), and detailed information of single-cell and pooled data is given in Tables EV1 and EV2. Briefly, reads were mapped to the mm9 genome using *hiclib* (which applies iterative mapping with *bowtie2*) and then filtered. These data were then converted into *Cooler* files with heatmaps at different resolutions for downstream analysis.

We applied the same methods for quantification of different features of spatial organization of the genome as done previously (Flyamer *et al*, 2017). Briefly, we used GC content as a proxy for A/B compartmentalization signal and constructed 5 × 5 percentile-binned matrices to quantify strength of compartment segregation (also called "saddle plots" for compartments). These 5 × 5 matrices were then iteratively corrected (Imakaev *et al*, 2012). For average analysis of TADs, we used published TAD coordinates (Rao *et al*, 2014) for the CH12-LX mouse cell line. We averaged Hi-C maps of all TADs and their neighboring regions, chosen to be of the same length as the TAD, after rescaling each TAD to a 90 × 90 matrix. For visualization, the contact probability of these matrices was rescaled to follow a shallow power law with distance (−0.25 scaling) (see Appendix Fig S1A). Similarly, we analyzed loops by summing up snHi-C contact frequencies for loop coordinates identified in Rao *et al* (2014) for CH12-LX mouse cells. By averaging 20 × 20 matrices surrounding the loops and dividing the final result by similarly averaged control matrices, we removed the effects of distance dependence (see Appendix Fig S1A). Control loop matrices were obtained by averaging 20 × 20 matrices centered on the locations of randomly shifted positions of known loops; shifts ranged from 100 to 1,100 kb with 100 shifts for each loop. For display and visual consistency with the loop strength quantification, we set the background levels of interaction to 1; the background is defined as the green boxes in Fig EV3A described below.

For the quantification of loop strength, we divided the average signal in the middle 6 × 6 submatrix by the average signal in top-left and bottom-right (at the same distance from the main diagonal) 6 × 6 submatrices (see Fig EV3A). To obtain the 95% confidence intervals on the loop strengths, we applied bootstrapping: Using the pooled single-cell data, we randomly sampled *N* loops with replacement (where *N* equals the total number of loops used in the original samples) and calculated the loop strengths from this random sample. We performed this procedure 10,000 times for each condition, using the sorted set of 10,000 strength values to obtain the confidence intervals. To test for significant differences between mean loop strengths between any two conditions, we used a permutation test. We calculated the mean loop strength for each pair of conditions being tested. Then, we calculated differences in mean loop strengths for data where the labels on replicates have been randomly permuted. We repeated the random permutation procedure 1,000 times and calculated *P*-values based on how frequently the "true" difference in loop strength was less than the difference in permuted data loop strengths.

Topologically Associating Domain strength was quantified using $P_c(s)$ normalized snHi-C data (see Appendix Fig S1A, bottom-left panel). In python notation, if M is the 90 × 90 TAD numpy array (where numpy is np) and L = 90 is the length of the matrix, then TAD_strength = box1/box2, where:

$$\text{box1} = 0.5 * \text{np.sum}(M[0:L//3, L//3:2*L//3]) \\ + 0.5 * \text{np.sum}(M[L//3:2*L//3, 2*L//3:L])$$

$$\text{box2} = \text{np.sum}(M[L//3:2*L//3, L//3:2*L//3]).$$

Compartment saddle plot strength was quantified by the formula: log(AA*BB/(AB*BA)), where AA, AB, BA, and BB represent the four corners of the iteratively corrected saddle plot matrix.

To calculate the insulation score, we computed the sum of read counts within a sliding 40-kb-by-40-kb diamond. The diamond was positioned such that the "tip" touched the main axis of the snHi-C map corresponding to a "self-interaction". Since snHi-C maps are not iteratively corrected, we normalized all insulation profiles by the score of the minimum insulation and then subtracted 1. This way, the insulation/domain boundary is at 0 and has a minimum of 0.

Contact probability ($P_c(s)$) curves were computed from 10-kb binned snHi-C data. We divided the linear genomic separations into logarithmic bins with a factor of 1.3. Data within these log-spaced bins (at distance, s) were averaged to produce the value of $P_c(s)$. In Fig 3, both $P_c(s)$ curves and their log-space slopes are shown following a Gaussian smoothing (using the scipy.ndimage.filters.gaussian_smoothing1d function with radius 0.8). Both the y-axis (i.e., log($P_c(s)$)) and the x-axis (i.e., log(s)) were smoothed.

### De novo TAD boundary calling

Topologically Associating Domain boundaries were called *de novo* on multiple cell types using the corner score as described in Schwarzer *et al* (2017) using default parameters. Hi-C data for this analysis were processed using *hiclib* as described (Imakaev *et al*, 2012), and files were converted to *Cool* format.

### Sorting maternal and paternal cells

As described previously (Flyamer *et al*, 2017), it is possible to distinguish maternally and paternally derived chromatin based on the shape of the $P_c(s)$ curve in single cells. Maternal chromatin has a characteristic plateau/flattening of the $P_c(s)$ at 10–30 Mb. Due to the similar pronucleus sizes of the Scc1 control data which made them difficult to sort post-lysis, we opted to sort maternal and paternal pronuclei *in silico*. We chose a separate cutoff value for $P_c(s)$ for G1-phase (Scc1 controls and knockout) and G2-phase cells (G2, Wapl controls and knockout) that was used to designate maternal or paternal chromatin as the $P_c(s)$ curve changes through the cell cycle. First, we normalized all $P_c(s)$ curves to 1 at 9 kb for all conditions. For G1 cells, all $P_c(s)$ curves with a value above $1 \times 10^{-4}$ at 15 Mb were designated maternal. For G2 cells, all $P_c(s)$ curves with a value above $2.5 \times 10^{-5}$ at 20 Mb were designated maternal. Cells in which the pronuclei were stuck together after lysis were given the tag "both" and were not assigned a maternal/paternal value *in silico*, but were used in the "combined" data analyses of Fig 2. We further filtered out bad data using the cutoff of $P_c(s) < 10^{-1}$ at 30 kb; these cells were excluded from all analyses.

### Analysis of Du *et al* (2017) data

Preprocessed, mapped valid pair files were obtained from GEO accession number GSE82185. These files were directly converted to the *Cooler* format (https://github.com/mirnylab/cooler) without any further filtering or processing using csort and cload functions. Averaging analysis for loops, TADs, and compartments was performed as described previously (Flyamer *et al*, 2017) and is summarized in the above section.

### Analyses of Ke *et al* (2017) data

FASTQ files were downloaded from BioProject, identifier PRJCA000241 (http://bigd.big.ac.cn/bioproject/browse/PRJCA000241). Data were mapped to the mm9 genome and converted to *Cooler* format using distiller (https://github.com/mirnylab/distiller-nf). Averaging analyses for loops, TADs, and compartments were performed as described previously (Flyamer *et al*, 2017).

### Polymer simulations

Polymer simulations of loop extrusion were performed as in Flyamer *et al* (2017), but using updates to the simulation engine (Fudenberg & Imakaev, 2017). The simulation engine is built using the openmm-polymer package which relies on OpenMM 7 (Eastman *et al*, 2017). Parameters for simulations were as follows: 2,000 MD steps per loop extrusion step. Simulations were performed using either $N = 30,000$ monomers or $N = 100,000$ monomers. Simulations were initialized using a fractal globule or a mitotic chromosome model, as described in Flyamer *et al* (2017). Bidirectional TAD boundaries were placed at monomers 0, 1,200, 1,500, 2,000, 2,900, 3,900, 4,300, 4,800, 5,600, 6,100, 6,500, 7,600, 8,300, 8,900, and 9,500 and at positions shifted by multiples of 10,000 (10,000, 11,200, 11,500, 12,000, ... 20,000, 21,200, 21,500, 22,000...). TAD boundaries were implemented as monomers that pause the loop-extruding factor (LEF) translocation with probability 99.5%. That would delay translocation of a LEF by on average 200 loop extrusion steps. All simulations were performed in periodic boundary conditions at a given density. For each simulation, we simulated 4,000 steps of loop extrusion dynamics, starting with a random placement of LEFs at the beginning of a simulation.

We performed two types of simulations. A parameter sweep for processivity–separation values was performed for a system of 30,000 monomers for all pairwise combinations of the values of processivity of 60, 120, 240, 480, and 960 kb and the values of separation of 30, 60, 120, 240, 480, and 2,400 kb. The largest value of separation was to simulate 20-fold depletion of LEFs relative to the wildtype model value of 120 kb (Fudenberg *et al*, 2016). All simulations here were initialized with a 30,000 monomer fragment of a mitotic chromosome model. We used a density of 0.02 for these simulations.

A more complete simulation was performed using a system of 100,000 monomers, initialized from a mitotic chromosome model, or from a fractal globule for maternal and paternal chromosomes, respectively. Particular values of parameters were chosen based on the parameter sweep. We chose values of processivity and separation of 120 kb for the control conditions model, the same values as used in Fudenberg *et al* (2016). For the model of *Scc1*[4], we reduced the number of cohesins 20-fold, which corresponds to increasing separation to 2,400 kb. For the model of *Wapl*[4] of maternal chromatin, we increased processivity fourfold, but kept the separation at 120 kb. For *Wapl*[4] of paternal chromatin, we best matched the difference in $P_c(s)$ in the s = 100–500 kb region by decreasing the processivity twofold, but increasing separation by twofold as compared to maternal. Additionally, to reflect the larger paternal pronuclear volume, we decreased the density of simulations twofold, to 0.01.

We calculated P$_c$(s) and simulated contact maps using a contact radius of 5 monomers. Both P$_c$(s) curves and their log-space slopes are shown following a Gaussian smoothing (using the scipy. ndimage.filters.gaussian_smoothing1d function with radius 0.8). Both the *y*-axis (i.e., log(P$_c$(s)) and the *x*-axis (i.e., log(s)) were smoothed.

**Data and software availability**

The snHi-C data have been deposited on NCBI GEO under the accession number GSE100569. Polymer simulation code is available in the "examples" directory of the openmm-polymer library (https:// bitbucket.org/mirnylab/openmm-polymer); analysis code of polymer configurations, including the surface area and volume measurements, will be made available upon publication. snHi-C data processing code has been released as an example for the hiclib package (https://bitbucket.org/mirnylab/hiclib).

**Expanded View** for this article is available online.

## Acknowledgements
We thank M. Coelho Correia da Silva and M. H. Idarraga-Amado for providing conditional *Wapl* mice and reagents, and A. Hirsch, K. Klien, and N. Laumann-Lipp for technical assistance with pronuclear extractions. We are grateful to J. Nuebler for discussions and suggestions for simulations and sharing codes, and to A. Goloborodko for suggesting the derivative-based analyses of P$_c$(s) curves. Illumina sequencing was performed at the VBCF NGS Unit (http://www.vbcf.ac.at) and Edinburgh Wellcome Trust Clinical Research Facility (WTCRF). We thank the staff of Vienna Biocenter BioOptics facility for assistance with imaging and analysis. J.G. is an associated student of the DK Chromosome Dynamics (W1238-B20) supported by the Austrian Science Fund (FWF) and the European Research Council. H.B.B. is grateful for support by the Natural Sciences and Engineering Research Council of Canada, PGS-D. I.M.F. is grateful for support from the Darwin Trust of Edinburgh. W.A.B. is supported by the UK Medical Research Council. This work was funded by the Austrian Academy of Sciences and by the ERC-StG-336460 ChromHeritance grant to K.T.-K. The work in the Mirny Laboratory is supported by R01 GM114190, U54 DK107980 from the National Institute of Health, and 1504942 from the National Science Foundation.

## Author contributions
JG and HBB contributed equally to the study. KT conceived the project. JG supervised by KT performed snHi-C and imaging experiments. HBB supervised by LAM developed and performed snHi-C data analysis and simulations. MI performed simulations. IMF performed image analysis. SL provided novel mouse strains and imaging reagents. J-MP provided a novel mouse strain. HBB, JG, MI, and IMF prepared the figures. HBB, JG, MI, IMF, WAB, LAM, and KT wrote the manuscript with input from all authors.

## Conflict of interest
The authors declare that they have no conflict of interest.

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
