## [Review Process File · The EMBO Journal]

A Mechanism of Cohesin-Dependent Loop Extrusion Organizes Zygotic Genome Architecture

Johanna Gassler, Hugo B. Brandão, Maxim Imakaev, Ilya M. Flyamer, Sabrina Ladstätter, Wendy A. Bickmore, Jan-Michael Peters, Leonid A. Mirny & Kikuë Tachibana

Review timeline:

Submission date:	24 August 2017
Editorial Decision:	26 September 2017
Revision received:	25 October 2017
Editorial Decision:	3 November 2017
Revision received:	6 November 2017
Accepted:	7 November 2017

Editor: Anne Nielsen

Transaction Report:

1st Editorial Decision

26 September 2017

Thank you again for submitting your manuscript for consideration by the EMBO Journal. It has now been seen by three referees whose comments are shown below.

As you will see from the reports, all referees express interest in the findings reported in your manuscript, although they also raise a number of points - mainly related to data presentation and analysis - that you will have to address this before they can support publication in The EMBO Journal.

For the revised manuscript I would particularly ask you to focus your efforts on the following points:

- > Elaborate on the statistics of the single cell analysis (nature of replicates, tests used, pooled experiments) as requested by refs #2 and #3
- > Refs #2 and #3 also both find that the averaging strategy used to compensate for the low amount of material could potentially lead to biased conclusions. I would therefore ask you to discuss this point and incorporate the suggestions made by the referees.
- > Another key point, discussed by both the refs and by yourself in the study, is the discrepancy (and re-analysis) with the data reported by Du et al. While we will not insist on increasing the number of biological replicates (as suggested by ref #2), you should provide additional description and discussion of how your analysis of the data from Du et al leads to a significantly different conclusion than that reported by the authors of that study.
- > As a final major point, ref #3 asks you to rephrase the description of the polymer analysis to make that part of the manuscript more accessible for the non-specialist reader.

Given the referees' overall positive recommendations, I would like to invite you to submit a revised version of the manuscript, addressing the comments of all three reviewers.

Thank you for the opportunity to consider your work for publication. I look forward to your

revision.

REFEREE REPORTS

Referee #1:

The Gassler et al. manuscript investigates genome organization by cohesin. The authors show that cohesin is required for the formation and/or maintenance of loops. Through the use of Wapl-deficient cells they provide evidence that cohesin structures the genome by loop extrusion, and the data are further supported by polymer simulations. The Wapl results are in line with recent published work, but the current paper adds key new insights by using mouse zygotes as a model system. The finding that DNA loops in maternal and paternal genomes are differentially affected by Wapl deficiency is fascinating, and the observation that cohesin is essential for loops is a major finding by itself.

I would support rapid publication of this manuscript with minor textual changes:

- 1) Page 4: the authors assume that the Nipbl/Mau2 complex is only required for cohesin loading. A recent preprint (Rhodes, bioRxiv) actually proposes that Nipbl also serves a purpose post cohesin loading, and Lopez-Serra (Nat Genet, 2014) suggest that the loader complex also plays a role in nucleosome remodelling. A simple solution would be to remove this half-sentence ("... and it is assumed that cohesin loading is the sole function of Nipbl/Mau2.")
- 2) Page 11, first sentence: The authors state that they demonstrate that cohesin is directly involved in forming loops and TADs. I am not quite convinced that they really show this. To my understanding, cohesin could also merely be involved in the maintenance of loops.
- 3) Figure 4B and C: Including the genotypes in the figure itself would be helpful to the reader. Now the genotypes are only in the legend.
- 4) Supplementary Figure 3, Figure heading: The acronym 'Pc(s)' will not be known to all readers. Best to spell it out here.
- 5) Supplementary Figure 3B, legend: Haarhuis et al. use quantitative immunofluorescence rather than quantitative western blots.

Referee #2:

The manuscript by Gassler et al. addresses the question of how and when chromatin conformations are established during embryonic development using a combination of single nucleus Hi-C and live cell imaging of zygotes/ early embryos of wild-type and KO mice, and polymer simulations. This study is a follow-up of their recently published report on chromatin organization in zygotes (Flyamer et al., 2017). The authors describe the dynamics of topologically associating domains (TADs), loops and compartments in early embryos, which they report are already present in zygotes and depend on cohesin, while Wapl affects loop and domain formation consistent with the loop extrusion model. Further, the authors suggest that differences in loop extrusion affect maternal and paternal chromatin compaction, respectively. Finally, the authors developed a model based on polymer simulations that suggests that cohesin's processivity is controlled by Wapl.

This study combines different techniques to convincingly provide mechanistic insight into the role of cohesin and Wapl in TAD and loop formation. Overall, the conclusions are justified by the data and should be of major interest for the field. However, there are several issues that need to be addressed:

Major concerns:

1. There is no information provided on how many nuclei were used for the presented data of snHi-C; do they represent biological and/or technical replicates? Statistical tests that specify if the reported differences are significant are missing for several experiments (e.g. Fig. 1D, 5C, S1B). How many snHi-C datasets were used for figures showing pooled data?
2. The authors re-analyzed data from Du et al., 2017 and concluded from this that loops and TADs are already present in early embryos, which opposes the conclusions drawn by Du et al. who claim that early embryos do not contain loops or TADs. It is therefore necessary to elaborate on this more comprehensively in the results and discussion sections. At the moment, it is difficult for the reader to follow how different analyses of the same dataset result in opposite results/conclusions and why the presented analysis is superior over the original conclusions by Du et al. In this regard, it would be also beneficial to explain, why in particular the list of loci from the CH-LX12 datasets from Rao et al., 2014 was used for this analysis (are there other Hi-C datasets with sufficient read depth that could have been used?).
3. The authors should explain in more detail why averaging is suitable for this analysis, and what types of controls have been carried out to test if the averaging leads to artefacts. They should also show control data (with random shifts of the Rao et al. 2014 loci) for the averaging for both TADs and loops. Also, can the authors please comment on whether they can estimate if their averaging results rather suggest a general tendency for loops to form in certain regions or if they demonstrate the actual existence of loops in a significant fraction of cells? If so can they give an estimate of the fraction of cells and prevalence of loops in different developmental stages?
4. Zp3-Wapl KO: There is no data presented confirming the successful KO of Wapl (as it was done for Scc1).
5. The authors raise the possibility that the loss of loops and domains in the absence of cohesin could be due to an indirect effect (e.g. on gene expression). To test this, they increased the residence time of cohesin on chromosomes by Wapl depletion. Although less likely, it could still be that the observed phenotype (stronger loops and TADs) by Wapl depletion could also be due to secondary effects (of Wapl depletion) on gene expression?
6. Live cell imaging (Fig.4): The authors detect Scc1-EGFP-vermicelli in Scc1-KO/Wapl-KO embryos but not in WT embryos. The resulting conclusion is that in the Wapl-KO background cohesin is not released, resulting in vermicelli formation. It would be beneficial, however, to repeat this experiment in Wapl-KO (only) embryos, to exclude the possibility that the Scc1-KO has an effect on this phenotype (also because the follow up analysis in Fig5 is done in Wapl-KO embryos). In addition, I fail to see a striking difference in vermicelli-morphology between the maternal and paternal chromatin. Quantification is needed to support this conclusion.
7. In Fig. 5, the authors analyze DNA morphology in fixed zygotes and state that 'additional DAPI-intense structures were visible specifically in the maternal nucleus (n=19/23).' However, as far as I can see, quantitative data is not provided.
8. The final section of the results describes observations made by polymer simulations and refers to 'data not shown'. This data needs to be presented. Also, based on the simulations, the authors conclude that loop extrusion by cohesin 'fosters inter-chromosomal interaction'. Since these simulations cannot be experimentally verified, the conclusion needs to be weakened - especially in the light of their simulations to confirm the microscopy data, in which the simulations predicted a different outcome than the experimental data presented.

Minor concerns:

1. Data shown in the figures should be described in more detail in the results section (an example is figure 1B: while the results on TADs and loops are briefly mentioned in the main text, the data on compartments is not described at all).

2. The orange color indicating the localization of cohesin in figure 3G-I and S3 is hard to see.
3. On pages 8/9, wild-type zygotes are only represented in figure 4B and S4A (not 4C, S4B), Scc1delta/Wapldelta zygotes only in 4C, S4B (not 4B, S4A).

Referee #3:

In this manuscript, the authors perform analysis of single-cell Hi-C as well as a re-analysis of data from recent low input population-based Hi-C experiments to examine the level of 3D chromatin structure in early mouse embryos. Briefly, in Fig. 1, the authors use aggregate loop, TAD and compartment analysis based on the low-input Hi-C data from (Du et al., Nature 2017), demonstrating that some level of structure can be distinguished in those datasets using this type of analysis. In addition, the authors also produced new single-cell Hi-C data for zygotes at G2 characterising 3D chromatin organisation at that stage. In Fig. 2, the authors present new single-cell Hi-C data for zygotes depleted of cohesin (delta Scc1) or enriched in cohesin by depleting Wapl (a factor responsible for cohesin unloading from chromatin). The authors use these data to characterise the changes mediated by the loss of cohesin (lack of loops and TADs and maintenance of compartments) or the loss of Wapl (increased loop strength and TAD formation), and to characterise the extent to which loops form in the mutants. These results are in agreement with recent reports characterising the effect of the loss of Wapl in bulk Hi-C (Haarhuis et al., Cell 2017) and with a pre-print looking at the effect of the loss of cohesin in bulk Hi-C (Rao et al., 2017). The authors then perform a comparison of the contact probability of control and cohesin or Wapl depleted embryos and develop a method to determine loop size and cohesin density from the contact probability curves (Fig 3). In Fig. 4 the authors perform time-lapse microscopy on GFP tagged Scc1 on Wapl depleted zygotes and observe the formation of vermicelli chromosomes, which seem to be more prominent in the maternal zygote. Since these embryos have not undergone zygotic genome activation at this stage, this suggests that the compaction observed is transcription independent. In Fig. 5, the authors perform image analysis on fixed embryos to characterise differences in compaction and homogeneity between the maternal and paternal pro-nucleus. Finally, in Fig. 6, the authors identify that the cohesin-depleted zygotes have a higher degree of inter-chromosomal contacts. The authors then perform polymer simulations with different parameters to try and find conformations that explain the observed increase in trans interactions, proposing that a diminish in the number of extruding loops would lead to chromosome surface roughening. Overall the manuscript is interesting and adds to the increasing amount of evidence characterising cohesin as a main regulator of chromatin conformation. The manuscript also tries to tackle a discrepancy between the authors previous work using single-cell Hi-C (Flyamer et al., Nature 2017), where they claim that TADs and loops are already present at the zygote stage, and (Du et al., Nature 2017; and Ke et al., Cell 2017) that use low-input bulk Hi-C to claim that zygotes do not have particularly strong conformational features. It is therefore important to sort out the discrepancy between these results.

Major points:

1. Aggregate analysis and limitations of single-cell Hi-C

Given the generally low amount of resulting contacts in single-cell Hi-C experiments, the authors resort to using aggregate analysis of loops, TADs and compartments to perform their analysis. Although I understand why the authors do that, I cannot but wonder whether aggregating these features over the whole genome (especially since these are done on uncorrected contact matrices) introduces specific biases that become apparent on the aggregate profiles. For example, would local insulation at TAD borders (for example through the binding of a chromatin factor, which might not generate a TAD per se, but that would result in local insulation) result in aggregate TAD plots similar to what the authors obtain in Fig 1B, C? The authors could test this by building simulated matrices with contact decays similarly as those described in (Du et al., Nature 2017) and introducing slight levels of insulation at TAD borders. If these simulations result in plots similar to those presented in Fig 1 for aggregate TADs or loops, that might identify local insulation as a confounding element of this analysis.

In addition to the point above, the authors show in this manuscript that early embryos display a significantly decreased level of chromatin structure when compared with older embryos (eg., at 8-cell stage). This point is in agreement in between this manuscript and (Du et al., Nature 2017; and

Ke et al., Cell 2017) and should be stressed in the manuscript. Given the potential limitations of the aggregate TAD and loop analysis, an alternative approach to finally settle the dispute would be to produce more biological replicates for the single-cell Hi-C data, that might allow a better characterisation of the contacts present in those samples as in (Nagano et al., Nature 2017) or using GAM (Beagrie et al., Nature 2017). I realise that this is a significant endeavour, and I am not suggesting that the authors perform thousands of single-cell Hi-C experiments, but given that TADs or loops are not visible on bulk Hi-C data for zygotes, a more comprehensive dataset (or one generated through an orthogonal technique, such as GAM) might be the only way to properly sort out this dispute.

Please note that the authors refer to the paucity of material as potential source for the lack of TADs in early zygotes. This is however unlikely since the ICM data presented in the same publication (Du et al., Nature 2017), where TADs can clearly be seen, was generated with an even lower number of cells as input material.

Please include single-cell Hi-C statistics. How many single-cell experiments were performed in total for each sample? How many contacts are detected in each library? Are these numbers comparable in G1/S G2 embryos?

2. Analysis of contact probabilities and modelling

I find the analysis of contact probabilities and the differences in between the samples that the authors refer to difficult to understand and the subsequent analyses of simulations difficult to follow. I think the analysis method might be very useful for comparing other samples, so this section should be clarified so a less specialised audience can understand it. In addition, the authors refer to a discrepancy between the average loop size obtained from their simulations and the features observed in Hi-C data (~1Mb). The authors offer an explanation based on the proposed stochastic nature of these loops, but I still do not understand why the simulations wouldn't converge at positions where the loops stall at full length.

In addition, there is a further discrepancy between the polymer simulation results and the vermicelli chromosomes, in particular for the maternal chromatin, which does not agree with the microscopy observations. This highlights a potential limitation of the polymer modelling approach. The authors acknowledge this fact and speculate that there might other mechanisms on top of the proposed loop extrusion that might play a role in the formation of vermicelli chromosomes. While this might be a reasonable strategy, the potential limitation of the current polymer simulation approach impacts the results in Fig 6, since no orthogonal data are presented to back up the conclusions on the chromosome surface roughening. Fig 6B in its current form is also very difficult to understand.

1st Revision - authors' response

25 October 2017

Responses to reviewers

Editor Summary

Editor:

For the revised manuscript I would particularly ask you to focus your efforts on the following points:

1. -> *Elaborate on the statistics of the single cell analysis (nature of replicates, tests used, pooled experiments) as requested by refs #2 and #3*

We have provided extensive information on the number of nuclei per experiment and number of nuclei used in pooled experiments in **new Table EV1 and EV2**. Due to the nature of the single-nucleus Hi-C protocol, each nucleus represents an independent biological replicate. We updated figure legends, inserted references to statistical tests in the text (e.g. **Figure 2D, Appendix Figure S1C, Figure 3D, Figure 7A**), and described the statistical tests in the methods section or named common tests in the main text. Tests used include Mann-Whitney U-test, and bootstrapping to obtain an estimate of the variation within a sample.

2. -> Refs #2 and #3 also both find that the averaging strategy used to compensate for the low amount of material could potentially lead to biased conclusions. I would therefore ask you to discuss this point and incorporate the suggestions made by the referees.

We have tested whether using TADs called de novo in many diverse cell types including inner cell mass cells of blastocyst embryos used by Du et al., 2017 and ES cells by Nora et al., 2017 show a contact enrichment in our zygote data set and in other cell types (**Figure 2C and Figure EV 1**). Notably, no contact enrichment is detected either in metaphase II oocytes (Du et al., 2017) with condensed chromosomes as expected of “mitotic-like” chromosomes (Naumova et al., 2013) or in cohesin knock-out zygotes (this work). TADs called from all cell types including CH-LX12 that we used originally, lead to a contact enrichment in zygotes. Interestingly, the contact enrichment in our zygote data is stronger than that of either Du et al. or Ke et al. zygotes, suggesting that snHi-C either works remarkably well for capturing contacts or has a higher signal to noise ratio compared to bulk Hi-C.

3. -> Another key point, discussed by both the refs and by yourself in the study, is the discrepancy (and re-analysis) with the data reported by Du et al. While we will not insist on increasing the number of biological replicates (as suggested by ref #2), you should provide additional description and discussion of how your analysis of the data from Du et al leads to a significantly different conclusion than that reported by the authors of that study.

Regarding how our analysis differs from that performed in Du et al.: It appears that TADs called from inner cell mass (ICM) cells of blastocysts were not rescaled (Du et al., 2017), which can lead to blurring of signal in aggregate analysis. This step is necessary to depict the structure of the TAD body independent of TAD sizes upon averaging. When we apply our approach to the Du et al. and Ke et al. zygote data, this also reveals contact enrichment. Moreover, we also directly examined Hi-C heatmaps of Du et al. and Ke et al. and find that there are both genomic regions lacking domain structures and other loci showing domain structures, which we have included as examples in **new Figure EV 2**. Therefore independently of aggregate analysis, contact enrichments of domain structures exist also in bulk Hi-C zygote data. Finally, we have elaborated on the discrepancies in the main text. Notably, we describe this in detail in the Discussion section; the entire 5th to last paragraph is devoted solely to this topic.

4. -> As a final major point, ref #3 asks you to rephrase the description of the polymer analysis to make that part of the manuscript more accessible for the non-specialist reader.

We have edited the polymer analysis to make it more accessible for non-specialist readers. In addition to text changes, we included schematic figures including **new Figure 1** to illustrate the difference between extruded and Hi-C loops and **new Figure 8A** to provide a visual representation of surface smoothening and roughening due to changes in loop extrusion.

Referee comments

Referee #1:

The Gassler et al. manuscript investigates genome organization by cohesin. The authors show that cohesin is required for the formation and/or maintenance of loops. Through the use of Wapl-deficient cells they provide evidence that cohesin structures the genome by loop extrusion, and the data are further supported by polymer simulations. The Wapl results are in line with recent published work, but the current paper adds key new insights by using mouse zygotes as a model system. The finding that DNA loops in maternal and paternal genomes are differentially affected by Wapl deficiency is fascinating, and the observation that cohesin is essential for loops is a major finding by itself.

I would support rapid publication of this manuscript with minor textual changes:

1) Page 4: the authors assume that the Nipbl/Mau2 complex is only required for cohesin loading. A recent preprint (Rhodes, bioRxiv) actually proposes that Nipbl also serves a purpose post cohesin

loading, and Lopez-Serra (Nat Genet, 2014) suggest that the loader complex also plays a role in nucleosome remodelling. A simple solution would be to remove this half-sentence ("... and it is assumed that cohesin loading is the sole function of Nipbl/Mau2.")

We thank the reviewer for supporting publication of the manuscript. We have amended the text accordingly.

2) Page 11, first sentence: The authors state that they demonstrate that cohesin is directly involved in forming loops and TADs. I am not quite convinced that they really show this. To my understanding, cohesin could also merely be involved in the maintenance of loops.

We agree that our data does not show that cohesin is “directly” involved in forming loops and TADs. Since we are examining loop and TAD establishment on sperm chromatin, which is hitherto compacted by protamines, one possibility is that loops and TADs are established de novo on paternal chromatin. Furthermore, Rao et al. 2017 recently showed that cohesin is required for loop formation following auxin-mediated cohesin degradation and washout. Nevertheless, we have amended the text to “formation or maintenance” of loops.

3) Figure 4B and C: Including the genotypes in the figure itself would be helpful to the reader. Now the genotypes are only in the legend.

We agree and have amended the figure accordingly.

4) Supplementary Figure 3, Figure heading: The acronym 'Pc(s)' will not be known to all readers. Best to spell it out here.

We have changed this to **Contact frequency (Pc(s)) vs Genomic distance (s, bp)**.

5) Supplementary Figure 3B, legend: Haarhuis et al. use quantitative immunofluorescence rather than quantitative western blots.

Thank you for pointing this out. We have corrected this.

Referee #2:

The manuscript by Gassler et al. addresses the question of how and when chromatin conformations are established during embryonic development using a combination of single nucleus Hi-C and live cell imaging of zygotes/ early embryos of wild-type and KO mice, and polymer simulations. This study is a follow-up of their recently published report on chromatin organization in zygotes (Flyamer et al., 2017). The authors describe the dynamics of topologically associating domains (TADs), loops and compartments in early embryos, which they report are already present in zygotes and depend on cohesin, while Wapl affects loop and domain formation consistent with the loop extrusion model. Further, the authors suggest that differences in loop extrusion affect maternal and paternal chromatin compaction, respectively. Finally, the authors developed a model based on polymer simulations that suggests that cohesin's processivity is controlled by Wapl.

This study combines different techniques to convincingly provide mechanistic insight into the role of cohesin and Wapl in TAD and loop formation. Overall, the conclusions are justified by the data and should be of major interest for the field. However, there are several issues that need to be addressed:

Major concerns:

1. There is no information provided on how many nuclei were used for the presented data of snHi-C; do they represent biological and/or technical replicates? Statistical tests that specify if the reported differences are significant are missing for several experiments (e.g. Fig. 1D, 5C, S1B). How many snHi-C datasets were used for figures showing pooled data?

We thank the reviewer for pointing this out and have provided extensive information in **new Tables EV1 and EV2** for number of nuclei used per experiment and which datasets were used in pooled data. For example, we used in total 17 *Wapl*^Δ nuclei, 18 *Wapl*^{fl} nuclei, 45 *Scc1*^Δ nuclei and 30 *Scc1*^{fl} nuclei that each represent a biological replicate. We note that this cell number is not unusual for single-cell Hi-C experiments; notably Stevens et al., Nature 2017 used 8 cells to study 3D chromatin structures. We further updated the figure legends and the methods section to include information about the statistical tests used.

2. The authors re-analyzed data from Du et al., 2017 and concluded from this that loops and TADs are already present in early embryos, which opposes the conclusions drawn by Du et al. who claim that early embryos do not contain loops or TADs. It is therefore necessary to elaborate on this more comprehensively in the results and discussion sections.

We agree that this requires more explanation and have elaborated on this in the results, and devote the entire 5th to last paragraph of the discussion section to this point. For example, our analysis differs from that performed in Du et al.: It appears that TADs called from inner cell mass (ICM) cells of blastocysts were not rescaled (Du et al., 2017), which can lead to blurring of signal in aggregate analysis. This step is necessary to depict the structure of the TAD body independent of TAD sizes upon averaging. When we apply our approach to the Du et al. and Ke et al. zygote data, this also reveals contact enrichment. Moreover, we also directly examined Hi-C heatmaps of Du et al. and Ke et al. and find that there are both genomic regions lacking domain structures and other loci showing domain structures, which we have included as examples in new **Figure EV 2**. Therefore independently of aggregate analysis, contact enrichments of domain structures exist also in bulk Hi-C zygote data.

At the moment, it is difficult for the reader to follow how different analyses of the same dataset result in opposite results/conclusions and why the presented analysis is superior over the original conclusions by Du et al.

We included the new text in the Discussion section to explain differences in the treatment of data (5th last paragraph):

“We demonstrated that TADs and loops can be clearly identified at all embryonic development stages, when known positions of TADs and loops were used. To exclude biases introduced by TAD positions used in the analysis, we tested TADs identified in many diverse cell types as well as TADs called *de novo* in bulk Hi-C of inner cell mass cells of blastocyst embryos (Du et al., 2017). Our ability to detect TADs in re-analyses of bulk Hi-C studies (Du et al., 2017, Ke et al., 2017) that were unable to detect them at early stages can be attributed to the higher statistical power of methods that we employed: not only did we aggregate TADs from positions called in population Hi-C data, but we also used observed-over-expected maps to correct for $P_c(s)$ specific for the used Hi-C map and rescaled TADs of different sizes (100 kb-1 Mb), allowing to depict the structure of the TAD body independently of TAD sizes upon averaging. The lack of rescaling of TADs (as well as different normalization) in the original analysis could have lead to blurring of signal in aggregate analysis. Hi-C for mitotic cells serves as a negative control as these show no TAD structures. We further validated our method by visual inspection of Hi-C maps that showed both regions lacking contact enrichment and other regions containing domain structures. We furthermore show that the structures detected by aggregate analysis depend critically on cohesin, which is in line with its proposed role in loop and TAD formation.”

In this regard, it would be also beneficial to explain, why in particular the list of loci from the CH-LX12 datasets from Rao et al., 2014 was used for this analysis (are there other Hi-C datasets with sufficient read depth that could have been used?).

We agree with the reviewer and have used TADs called *de novo* in many diverse cell types including inner cell mass used by Du et al. 2017 and ES cells by Nora et al., 2017. This analysis shows a contact enrichment in our zygote data set and in other cell types (**new Figure 2C and Figure EV1**). Our analysis includes a number of controls; for example, no contact enrichment is detected either in metaphase II oocytes (Du et al) with condensed chromosomes as expected of “mitotic-like” chromosomes (Naumova et al., 2013) or in cohesin knock-out zygotes (this work). TADs called from all cell types including CH-LX12 that we used originally, lead to a contact

enrichment in zygotes. Interestingly, the contact enrichment in our zygote data is stronger than that of either Du et al or Ke et al zygotes, suggesting that snHi-C works remarkably well for capturing contacts or has a higher signal to noise ratio compared to bulk Hi-C.

In a nutshell, we have performed an extensive analysis that converges on contact enrichments for TADs in zygotes irrespective of the set of TADs used as a reference. In addition, cohesin knock-out abrogates contact enrichments whereas Wapl knock-out leads to an increased contact enrichment comparable to that of somatic cells. Therefore, the contact enrichment observed in our wild-type zygote data reflects chromatin structures that critically depend on cohesin.

3. The authors should explain in more detail why averaging is suitable for this analysis, and what types of controls have been carried out to test if the averaging leads to artefacts. They should also show control data (with random shifts of the Rao et al. 2014 loci) for the averaging for both TADs and loops. Also, can the authors please comment on whether they can estimate if their averaging results rather suggest a general tendency for loops to form in certain regions or if they demonstrate the actual existence of loops in a significant fraction of cells? If so can they give an estimate of the fraction of cells and prevalence of loops in different developmental stages?

The way loop strength is being established indeed uses the strategy proposed by the reviewer, i.e. by considering contact enrichment at annotated loops vs randomly shifted positions. For each loop, we used two controls: (i) positions randomly shifted along the same diagonal; (ii) controls positions shifted up and down the diagonal by 60 kb. The aggregate maps, centered at loop positions, were divided by the average signal from random shifting. The strength of the loop was further quantified as the ratio of the signal, in this normalized map, at the loop position and the mean of the two 60 kb shifted squares. This methodology is explained in **Appendix Figure S1A**.

We agree with the reviewer that it would be very informative to understand whether the enrichment at loops that we observe reflects a general tendency for loops or the actual existence of loops in a significant fraction of cells. To answer this question we examined the distribution of the number of contacts at positions of annotated loops, and at control (shifted positions). These distributions didn't show possible bimodality or heavy tails, as more than 80% of loop instances contain less than 5 reads. This suggests that existing read density was insufficient to identify clusters of contacts as existence of a specific loop in a specific cell.

Thus we can only see loops as enrichments of contacts in aggregate maps. Although we cannot identify frequency of individual loops, we know from cell population Hi-C that contact probability at loops is generally 3-100 times less than the contact probability between loci separated by a 10 kb (Fudenberg et al Cell Reports 2016), demonstrating that each loop is realized in a small fraction of cells.

4. Zp3-Wapl KO: There is no data presented confirming the successful KO of Wapl (as it was done for Scc1).

Genotyping of pups born to **Wapl** f/f *Zp3*-Cre females mated to wild-type males show >98% deletion efficiency (n=85 mice) (M. da Silva, J. M. Peters, personal communication), indicating a highly efficient knock-out of Wapl. To determine whether Wapl protein is efficiently depleted, we tested Wapl antibodies from Jan-Michael Peters lab but these could not detect Wapl by immunofluorescence. We therefore unfortunately cannot provide quantitative data on Wapl protein depletion but the vermicelli phenotype is fully penetrant in Wapl knock-out zygotes, suggesting that Wapl is depleted sufficiently to increase cohesin's residence time on chromatin.

5. The authors raise the possibility that the loss of loops and domains in the absence of cohesin could be due to an indirect effect (e.g. on gene expression). To test this, they increased the residence time of cohesin on chromosomes by Wapl depletion. Although less likely, it could still be that the observed phenotype (stronger loops and TADs) by Wapl depletion could also be due to secondary effects (of Wapl depletion) on gene expression?

We agree that formally we cannot exclude that indirect effects could alter chromatin structure. However, the model of cohesin-dependent loop extrusion predicts that cohesin loss leads to loss of loops and increasing cohesin residence time by Wapl depletion leads to extended loop extrusion and

thus stronger loops. The phenotypes observed are consistent with both predictions, suggesting that the loss of loops and domains in the absence of cohesin is most likely a direct effect. However, to indicate that the effect could still be indirect we added the following sentence: “Although formally we cannot exclude that these effects are due to changes in gene expression, the most parsimonious explanation for both loss of cohesin leading to loss of loops, and increase of cohesin residence time by Wapl depletion leading to stronger loops, is that the effect of cohesin on loops and TADs is direct.”

6. Live cell imaging (Fig.4): The authors detect Scc1-EGFP-vermicelli in Scc1-KO/Wapl-KO embryos but not in WT embryos. The resulting conclusion is that in the Wapl-KO background cohesin is not released, resulting in vermicelli formation. It would be beneficial, however, to repeat this experiment in Wapl-KO (only) embryos, to exclude the possibility that the Scc1-KO has an effect on this phenotype (also because the follow up analysis in Fig5 is done in Wapl-KO embryos). In addition, I fail to see a striking difference in vermicelli-morphology between the maternal and paternal chromatin. Quantification is needed to support this conclusion.

We have performed live-cell imaging of *Wapl*^Δ zygotes expressing Scc1-EGFP (**new Appendix Figure S4**) and find that we also observe vermicelli formation, excluding the possibility that *Scc1*^Δ has a major effect on the phenotype. The background signal of nuclear fluorescence is higher in this experiment, possibly because free Scc1-EGFP (lacking SMC heterodimers) might accumulate in the presence of endogenous Scc1. The morphological difference we observe is challenging to quantify in the live cells, and we therefore focused on extending our analysis of quantifying chromatin compaction (please see below).

7. In Fig. 5, the authors analyze DNA morphology in fixed zygotes and state that 'additional DAPI-intense structures were visible specifically in the maternal nucleus (n=19/23).' However, as far as I can see, quantitative data is not provided.

In addition to the grey-level co-occurrence matrix analysis performed previously, we have now performed an additional analysis of DAPI-intense structures in segmented nuclei. After thresholding the nuclei by DAPI intensity, we compared size distribution of identified objects between conditions and pronuclei. We show that the size of DAPI-intense structures significantly increases upon Wapl knock-out (p-values 1.25×10^{-11} and 8.23×10^{-28} for maternal and paternal pronuclei, respectively), and that the maternal pronuclei contain slightly bigger objects than paternal pronuclei (p-value 0.00014), consistent with stronger vermicelli (**Figure 6E and Appendix Figure S6B**).

8. The final section of the results describes observations made by polymer simulations and refers to 'data not shown'. This data needs to be presented. Also, based on the simulations, the authors conclude that loop extrusion by cohesin 'fosters inter-chromosomal interaction'. Since these simulations cannot be experimentally verified, the conclusion needs to be weakened - especially in the light of their simulations to confirm the microscopy data, in which the simulations predicted a different outcome than the experimental data presented.

We performed new simulations where we systematically swept parameters space and examined ability to reproduce “vermicelli” formed by loop-extruding cohesins. Indeed we found a range of parameters that leads to chromosomes that reproduce vermicelli phenotype observed in microscopy of *Wapl*^Δ zygotes (**Figure EV 4A and Appendix Figure S3**). We also extended our simulations of the effect of loop extrusion on the surface area of chromosomes, demonstrating that loop extrusion can affect chromosome surface area and hence trans- interactions, as observed in Hi-C (**Figure 7B, Figure 8 and Appendix Figure S9**). We revised this section of the manuscript.

Minor concerns:

- 1. Data shown in the figures should be described in more detail in the results section (an example is figure 1B: while the results on TADs and loops are briefly mentioned in the main text, the data on compartments is not described at all).*
- 2. The orange color indicating the localization of cohesin in figure 3G-I and S3 is hard to see.*
- 3. On pages 8/9, wild-type zygotes are only represented in figure 4B and S4A (not 4C, S4B), Scc1delta/Wapldelta zygotes only in 4C, S4B (not 4B, S4A).*

We appreciate these comments and made changes to make figures more consistent and easier to read. To make the localization of cohesin easier to see, we have included a new **Appendix Figure S3** that shows a 2D projection of cohesin locations for many different simulation parameters.

Referee #3:

In this manuscript, the authors perform analysis of single-cell Hi-C as well as a re-analysis of data from recent low input population-based Hi-C experiments to examine the level of 3D chromatin structure in early mouse embryos. Briefly, in Fig. 1, the authors use aggregate loop, TAD and compartment analysis based on the low-input Hi-C data from (Du et al., Nature 2017), demonstrating that some level of structure can be distinguished in those datasets using this type of analysis. In addition, the authors also produced new single-cell Hi-C data for zygotes at G2 characterising 3D chromatin organisation at that stage. In Fig. 2, the authors present new single-cell Hi-C data for zygotes depleted of cohesin (delta Scc1) or enriched in cohesin by depleting Wapl (a factor responsible for cohesin unloading from chromatin). The authors use these data to characterise the changes mediated by the loss of cohesin (lack of loops and TADs and maintenance of compartments) or the loss of Wapl (increased loop strength and TAD formation), and to characterise the extent to which loops form in the mutants. These results are in agreement with recent reports characterising the effect of the loss of Wapl in bulk Hi-C (Haarhuis et al., Cell 2017) and with a pre-print looking at the effect of the loss of cohesin in bulk Hi-C (Rao et al., 2017). The authors then perform a comparison of the contact probability of control and cohesin or Wapl depleted embryos and develop a method to determine loop size and cohesin density from the contact probability curves (Fig 3). In Fig. 4 the authors perform time-lapse microscopy on GFP tagged Scc1 on Wapl depleted zygotes and observe the formation of vermicelli chromosomes, which seem to be more prominent in the maternal zygote. Since these embryos have not undergone zygotic genome activation at this stage, this suggests that the compaction observed is transcription independent. In Fig. 5, the authors perform image analysis on fixed embryos to characterise differences in compaction and homogeneity between the maternal and paternal pro-nucleus. Finally, in Fig. 6, the authors identify that the cohesin-depleted zygotes have a higher degree of inter-chromosomal contacts. The authors then perform polymer simulations with different parameters to try and find conformations that explain the observed increase in trans interactions, proposing that a diminish in the number of extruding loops would lead to chromosome surface roughening. Overall the manuscript is interesting and adds to the increasing amount of evidence characterising cohesin as a main regulator of chromatin conformation. The manuscript also tries to tackle a discrepancy between the authors previous work using single-cell Hi-C (Flyamer et al., Nature 2017), where they claim that TADs and loops are already present at the zygote stage, and (Du et al., Nature 2017; and Ke et al., Cell 2017) that use low-input bulk Hi-C to claim that zygotes do not have particularly strong conformational features. It is therefore important to sort out the discrepancy between these results.

Major points:

1. Aggregate analysis and limitations of single-cell Hi-C

Given the generally low amount of resulting contacts in single-cell Hi-C experiments, the authors resort to using aggregate analysis of loops, TADs and compartments to perform their analysis. Although I understand why the authors do that, I cannot but wonder whether aggregating these features over the whole genome (especially since these are done on uncorrected contact matrices) introduces specific biases that become apparent on the aggregate profiles. For example, would local insulation at TAD borders (for example through the binding of a chromatin factor, which might not generate a TAD per se, but that would result in local insulation) result in aggregate TAD plots similar to what the authors obtain in Fig 1B, C? The authors could test this by building simulated matrices with contact decays similarly as those described in (Du et al., Nature 2017) and introducing slight levels of insulation at TAD borders. If these simulations result in plots similar to those presented in Fig 1 for aggregate TADs or loops, that might identify local insulation as a confounding element of this analysis.

We thank the reviewer for pointing this out. During the averaging procedure for loops we also average the coverage profiles to control for any biases that affect coverage (e.g. GC content, chromatin accessibility or restriction site density) according to the procedure we introduced previously (Flyamer et al., 2017). We also note, that Leonid Mirny and colleagues have previously

used polymer modelling to show that introduction of weak local insulation does not lead to formation of TADs (see Figure S5 in Fudenberg et al., 2016, Cell Reports) - this is due to the requirement for a long-distance insulation at TAD boundaries, consistent with the loop extrusion model.

However we have now extended our TAD analysis by testing whether using TADs called de novo in many diverse cell types including inner cell mass used by Du et al. 2017 and ES cells by Nora et al., 2017 show a contact enrichment in our zygote data set and in other cell types (**Figure 2C** and **Figure EV1**). Our analysis includes a number of controls; for example, no contact enrichment is detected either in metaphase II oocytes (Du et al) with condensed chromosomes as expected of “mitotic-like” chromosomes (Naumova et al., 2013) or in cohesin knock-out zygotes (this work), therefore the enrichment we observe does not come from technical biases. TADs called from all cell types including CH-LX12 that we used originally, lead to a contact enrichment in zygotes. Interestingly, the contact enrichment in our zygote data is stronger than that of either Du et al or Ke et al zygotes, suggesting that snHi-C works remarkably well for capturing contacts compared to bulk Hi-C.

Lastly, we also directly examined Hi-C heatmaps of Du et al. and Ke et al. and find that there are both genomic regions lacking domain structures and other loci showing domain structures, which we have included as examples in **new Figure EV2**. Therefore independently of aggregate analysis, contact enrichments of domain structures exist also in bulk Hi-C zygote data.

In addition to the point above, the authors show in this manuscript that early embryos display a significantly decreased level of chromatin structure when compared with older embryos (eg., at 8-cell stage). This point is in agreement in between this manuscript and (Du et al., Nature 2017; and Ke et al., Cell 2017) and should be stressed in the manuscript.

We thank the reviewer for pointing this out and have amended the text accordingly (pages 4 and 13, in Results and Discussion, respectively).

Given the potential limitations of the aggregate TAD and loop analysis, an alternative approach to finally settle the dispute would be to produce more biological replicates for the single-cell Hi-C data, that might allow a better characterisation of the contacts present in those samples as in (Nagano et al., Nature 2017) or using GAM (Beagrie et al., Nature 2017). I realise that this is a significant endeavour, and I am not suggesting that the authors perform thousands of single-cell Hi-C experiments, but given that TADs or loops are not visible on bulk Hi-C data for zygotes, a more comprehensive dataset (or one generated through an orthogonal technique, such as GAM) might be the only way to properly sort out this dispute.

Indeed the use of orthogonal methods as an independent means to show TADs and loops in zygotes would help to strengthen this point but we feel that applying these methods for the first time to embryos is beyond the scope of this study. Nevertheless, we have addressed some potential limitations of aggregate TAD and loop analysis by providing additional controls including random shifts of loops from Rao et al., 2014 (shown in **Figure S1A**), explaining possible discrepancies in the aggregate analysis between our work and Ke et al., (see discussion 5th last paragraph) and extended the aggregate analysis to TADs called de novo in a variety of cell types (**new Figure 2C** and **Figure EV1**), showing that these produce contact enrichments in all currently available zygote data (this work, Flyamer et al., 2017, Du et al., 2017, Ke et al., 2017).

Please note that the authors refer to the paucity of material as potential source for the lack of TADs in early zygotes. This is however unlikely since the ICM data presented in the same publication (Du et al., Nature 2017), where TADs can clearly be seen, was generated with an even lower number of cells as input material.

We agree that this is a somewhat confusing point. For Ke et al., 2238 zygotes and 66 blastocysts were used in the largest replicates (their Suppl Table). Blastocysts isolated on e3.5 consist of ~90 cells (Kang et al., Dev Cell 2017), suggesting that ~6000 cells were used for the blastocyst experiment. For Du et al., cell numbers as opposed to embryo numbers are cited and 260-468 zygotes and 570-1400 ICM cells were used (their Suppl Table); note that the cell numbers cited for ICM are in fact estimates (personal communication with Zhenhai Du, first author of the study).

Therefore the ICM data is derived from a larger cell number than the zygote data, which might explain the stronger detection of TADs in later stage embryos. Nevertheless, we agree with the conclusions of Du et al. and Ke et al. that zygotic TADs are likely weaker than TADs at later stage embryos.

Using TADs called in diverse cell types, we also find that contact enrichments in our zygote data set is higher than in the Ke et al. and Du et al. data sets (new **Figure 2C** and **Figure EV 1**), suggesting that contact capture is likely more efficient with snHi-C than bulk Hi-C approaches.

Please include single-cell Hi-C statistics. How many single-cell experiments were performed in total for each sample? How many contacts are detected in each library? Are these numbers comparable in G1/S G2 embryos?

We have provided the information in **new Tables EV1 and EV2** for number of nuclei used per experiment and number of nuclei used in pooled experiments including detected contacts per library. For example we used in total 17 *Wapl*^A nuclei, 18 *Wapl*^I nuclei, 45 *Sccl*^A nuclei and 30 *Sccl*^I nuclei that each represent a biological replicate. G1/S experiments consist of 31 maternal and 24 paternal nuclei, while G2 consist of 18 maternal and 13 paternal nuclei. As for number of contacts detected in each dataset, maternal G1/S experiments yielded 3,959,990 cis-contacts and paternal 2,627,375 cis-contacts. In G2 these numbers are 3,333,107 cis-contacts in maternal and 2,414,385 cis-contacts in paternal nuclei, making the data well comparable.

2. Analysis of contact probabilities and modelling

I find the analysis of contact probabilities and the differences in between the samples that the authors refer to difficult to understand and the subsequent analyses of simulations difficult to follow. I think the analysis method might be very useful for comparing other samples, so this section should be clarified so a less specialised audience can understand it. In addition, the authors refer to a discrepancy between the average loop size obtained from their simulations and the features observed in Hi-C data (~1Mb). The authors offer an explanation based on the proposed stochastic nature of these loops, but I still do not understand why the simulations wouldn't converge at positions where the loops stall at full length.

We revised this section of the manuscript and provide detailed and accessible explanation of our modeling strategy. We also emphasized the difference between extruded loops and peaks of contact frequency in Hi-C maps (that are frequently referred to as Hi-C loops). We also developed a cartoon explaining basic concepts of this analysis (**new Figure 1**).

In addition, there is a further discrepancy between the polymer simulation results and the vermicelli chromosomes, in particular for the maternal chromatin, which does not agree with the microscopy observations.

We performed new simulations where we systematically swept parameter space and examined ability to reproduce “vermicelli” formed by loop-extruding cohesins. Indeed we found a range of parameters that leads to chromosomes that reproduce vermicelli phenotype observed in microscopy of *Wapl^A* zygotes (**Appendix Figure S3**).

This highlights a potential limitation of the polymer modelling approach. The authors acknowledge this fact and speculate that there might other mechanisms on top of the proposed loop extrusion that might play a role in the formation of vermicelli chromosomes. While this might be a reasonable strategy, the potential limitation of the current polymer simulation approach impacts the results in Fig 6, since no orthogonal data are presented to back up the conclusions on the chromosome surface roughening. Fig 6B in its current form is also very difficult to understand.

We thank the reviewer for pointing this out. We have amended the text to indicate very clearly that this is a model that will require experimental validation, which we believe is presently beyond the scope of this manuscript. We further added a schematic figure, Figure 8A, to provide a visual representation of surface smoothing or roughening due to changes in loop extrusion.

2nd Editorial Decision

3 November 2017

Thank you for submitting a revised version of your manuscript. It has now been seen by one of the original referees and this person's comments are shown below. As you will see the referee finds that all criticisms have been sufficiently addressed and recommend the manuscript for publication. However, before we can go on to officially accept the manuscript there are a few editorial issues concerning text and figures that I need you to address in a final revision.

Thank you again for giving us the chance to consider your manuscript for The EMBO Journal, I look forward to receiving your final revision.

REFeree REPORT

Referee #3:

The authors have successfully addressed all my comments and I support publication of this manuscript.

YOU MUST COMPLETE ALL CELLS WITH A PINK BACKGROUND

Corresponding Author Name: Kikue Tachibana

Manuscript Number: EMBOJ-2017-98083